# How Do Diffusion Models Improve Adversarial Robustness?

## Abstract

Recent findings suggest that diffusion models significantly enhance empirical adversarial robustness. While some intuitive explanations have been proposed, the precise mechanisms underlying these improvements remain unclear. In this work, we systematically investigate how and how well do diffusion models improve adversarial robustness. First, we observe that diffusion models intriguingly increase—rather than decrease—the $\ell_p$ distances to clean samples. This is the opposite of what was believed previously. Second, we find that the purified images are heavily influenced by the internal randomness of diffusion models. To properly evaluate the robustness of systems with inherent randomness, we introduce the concept of fuzzy adversarial robustness, and find that empirically a substantial fraction of adversarial examples are fuzzy in nature. Finally, by leveraging a hyperspherical cap model of adversarial regions, we show that diffusion models increase robustness by dramatically compressing the image space. Our findings provide novel insights into the mechanisms behind the robustness improvements of diffusion model-based purification and offer guidance for the development of more efficient adversarial purification systems.

## 1 Introduction

Neural networks are vulnerable to small adversarial perturbations (Szegedy et al., 2013; Goodfellow et al., 2014). This presents a fundamental question on the robustness of artificial learning systems. Adversarial training (Madry et al., 2017) has been proposed as a successful method to overcome this problem (Shafahi et al., 2019; Pang et al., 2020; Wang et al., 2021). However, research has found that training with a specific attack usually sacrifices the robustness against other types of perturbations (Schott et al., 2018; Ford et al., 2019; Yin et al., 2019), indicating that adversarial training overfits the attack rather than achieving an overall robustness improvement.

Adversarial purification provides an alternative path toward adversarial robustness. This approach typically relies on generative models to purify the stimulus before passing to a classifier(Song et al., 2018; Samangouei et al., 2018; Shi et al., 2021; Yoon et al., 2021). The basic idea is to leverage the image priors learned in some generative models to project adversarial perturbations back toward the image manifold. Intuitively, the performance of such purification should depend on how well the generative models capture the probability distribution of natural images. Recently, adversarial purification based on diffusion models(Ho et al., 2020; Song et al., 2020b) (DiffPure) was reported to show promising improvements against various empirical attacks on multiple datasets (Nie et al., 2022). Such a direction of using diffusion models as denoisers was further combined with the deonise smoothing framework (Cohen et al., 2019; Salman et al., 2020) to improve certified robustness (Carlini et al., 2022; Xiao et al., 2023). However, more recent work (Lee & Kim, 2023) argued that there was an overestimate of the robustness improvement from the DiffPure method. In all, despite of some promising empirical results, the underlying mechanism of empirical robustness improvement from diffusion models (how), as well as a proper robustness evaluation with randomness (how well), were not yet well understood.

To close this important gap, we systematically investigate how diffusion models improve adversarial robustness. In this paper, we report a set of surprising phenomena of diffusion models, and furthermore identify the key mechanisms for robustness improvements under diffusion-model-based adversarial purification. Our main contributions are summarized below:

- First, we observe that, somewhat surprisingly, diffusion models increase—rather than decrease—the $\ell_p$ distances to clean samples (Sec. 3.1), with the purified states being heavily influenced by internal randomness (Sec. 3.2).

- To properly evaluate the robustness of systems with inherent randomness, we introduce the concept of fuzzy adversarial robustness (Sec. 4.1), and find that most of the traditionally believed adversarial examples are fuzzy in nature (Sec. 4.2).

- Using a hyperspherical cap model of adversarial regions (Sec. 5), we show that diffusion models increase adversarial robustness by compressing the image space (Sec. 6).

## 2 RELATED WORK AND PRELIMINARIES

**Generative models for adversarial purification**   Unlike adversarial training which directly augments the classifier training with adversarial attacks, adversarial purification intends to first "purify" the perturbed image before classification. Generative models are usually utilized as the purification system, such as denoising autoencoder (Gu & Rigazio, 2014), denoising U-Net (Liao et al., 2018), PixelCNN (Song et al., 2018) and GAN (Samangouei et al., 2018). Denote the purification system as $f(\boldsymbol{x})$, and the following readout classifier as $g(\boldsymbol{x})$. Under the assumption that adversarial purification processes perturbed data $\tilde{\boldsymbol{x}}$ close to the clean data $\boldsymbol{x}$, thus $f(\tilde{\boldsymbol{x}}) \approx \boldsymbol{x}$, the Bypass Direct Approximation (BPDA) (Athalye et al., 2018) can provide a robustness estimation if $f$ is hard-to-differentiate.

Diffusion models (Ho et al., 2020; Song et al., 2020b) set the SOTA performances on image generation, and represent a natural choice for adversarial purification. Nie et al. (2022) proposed the DiffPure framework, which utilized both the forward and reverse process and achieved promising empirical robustness comparable with adversarial training on multiple benchmarks. Similar improvements were reported with guided diffusion models (Wang et al., 2022). These studies led to substantial interest in applying diffusion models for adversarial purification in various domains, including auditory data (Wu et al., 2022) and 3D point clouds (Sun et al., 2023). Recently, other tricks such as adversarial guidance (Lin et al., 2024) were also introduced to further enhance robustness. However, through a comprehensive experimental evaluation, (Lee & Kim, 2023) discovered that there was an overestimate of the robustness improvement from diffusion models, and it was recommended to apply the PGD-EOT with full gradients directly over AutoAttack. Another line of research applies diffusion models to improve certified robustness Cohen et al. (2019). Carlini et al. (2022) found that plugging diffusion models as a denoiser into the denoised smoothing framework (Salman et al., 2020) can lead to non-trivial certified robustness. Xiao et al. (2023) further developed this method and explained the improvement in certified robustness.

**Diffusion models and randomness**   Diffusion models consist of forward diffusion and reverse denoising processes. The forward process of Denoising Diffusion Probabilistic Models (DDPM) (Ho et al., 2020) is

$$\boldsymbol{x}_t = \sqrt{\alpha_t}\boldsymbol{x}_{t-1} + \sqrt{1 - \alpha_t}\boldsymbol{\epsilon}, \ \ \boldsymbol{\epsilon} \sim \mathcal{N}(\boldsymbol{0}, \boldsymbol{I}), \tag{1}$$

in which the $\boldsymbol{\epsilon}$ will introduce randomness. Further, the reverse process

$$\boldsymbol{x}_{t-1} = \frac{1}{\sqrt{\alpha_t}}\left(\boldsymbol{x}_t - \frac{1 - \alpha_t}{\sqrt{1 - \bar{\alpha}_t}}\boldsymbol{\epsilon}_\theta(\boldsymbol{x}_t, t)\right) + \sigma_t \boldsymbol{z}, \tag{2}$$

also introduces randomness through $\boldsymbol{z} \sim \mathcal{N}(\boldsymbol{0}, \boldsymbol{I})$. Such randomness may raise concerns about gradient masking in robustness evaluation (Papernot et al., 2017), which provides a false sense of robustness against gradient-based attacks (Tramèr et al., 2018). Athalye et al. (2018) further identified that randomness could cause gradient masking as "stochastic gradients", and proposed the expectation-over-transformation (EOT) which became the standard evaluation for stochastic gradients (Carlini et al., 2019). How to properly understand the effect of randomness in robustness evaluation is controversial and still debatable (Gao et al., 2022; Yoon et al., 2021).

**Geometry of adversarial regions**   A line of studies has examined the geometry of adversarial spaces. Goodfellow et al. (2014) first showed that rather than being scattered droplets, adversarial examples form relatively large, continuous regions (Warde-Farley & Goodfellow, 2016). Tramèr et al. (2017) studied the dimensionality of the adversarial regions to explain the transferability of adversarial examples. Ma et al. (2018) further developed this direction with local intrinsic dimensions. Khoury & Hadfield-Menell (2018) highlighted that there were multiple off-manifold directions to

construct adversarial examples in a high dimensional space. The adversarial hyperspherical cap model we proposed here can be regarded as a direct extension to (Tramèr et al., 2017) and leads to an alternative proof of their GAAS arrangement (Appendix A.2).

## 3 INTRIGUING BEHAVIOR OF DIFFUSION MODELS

### 3.1 DIFFUSION MODELS PUSH THE PURIFIED IMAGES AWAY FROM CLEAN IMAGES

While the exact mechanisms for the robustness improvement under diffusion models remain unclear, intuitive explanations have been discussed in the DiffPure paper (Nie et al., 2022), e.g. diffusion models "recover clean images through the reverse denoising process". This motivates us to test a simple hypothesis, that is, diffusion models shrink the $\ell_p$ distances towards clean images during adversarial purification. Ideally, if a purification system is able to shrink the $\ell_p$ distances across multiple noise levels, we can apply the purification recursively to essentially turn the original adversarial attack into attacks with smaller magnitudes, thereby increasing robustness (see Fig. 1a). Below we will systematically examine this hypothesis with experiments.

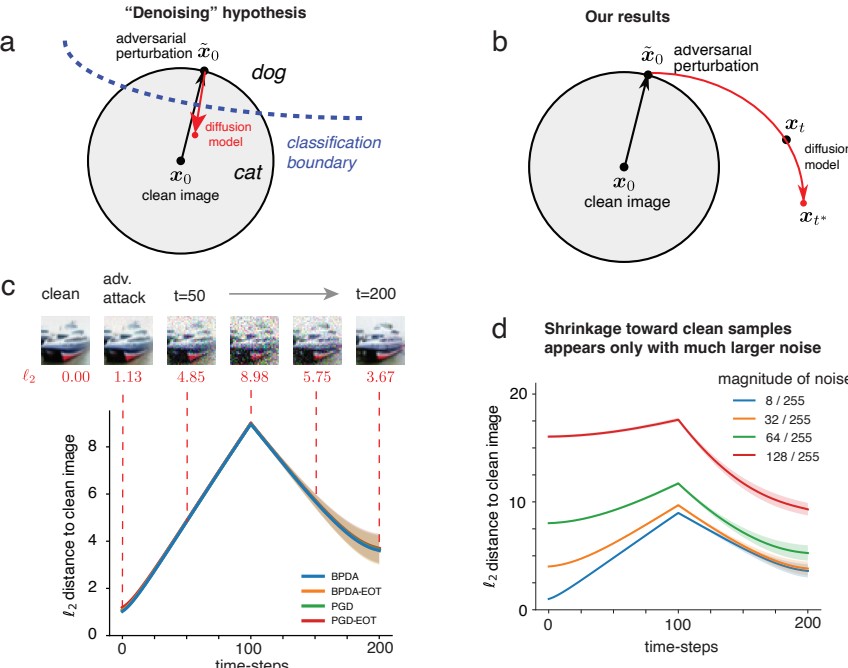

Figure 1: **Diffusion models do not shrinkage the distance to clean samples under adversarial purification.** (a) Schematic showing a population hypothesis on how diffusion models improve robustness by shrinkage towards clean samples. (b) Schematic summarizing our results, suggesting that the shrinkage hypothesis is not supported. (c) The $\ell_2$ distance measurements w.r.t. the clean images on CIFAR-10 during purification steps. Distances between the purified states $x_t$ and clean stimuli $x_0$ are measured with adversarial attacks as initial perturbations ($\ell_\infty = 8/255$). All models purify to states with larger $\ell_2$ distances, thus further away from the initial state. (d) $\ell_2$ distance measurements for different levels of uniform noise perturbations.

**No shrinkage of the adversarial attack after being transformed by diffusion models.** To test whether diffusion models shrink the distance of the adversarial attack to the clean image, we performed a series of numerical experiments. From a given clean image, we first generated an adversarial attack. We then fed the adversarial image through the diffusion model, and quantified the distance between the purified states and the clean images. See Fig. 1 c for an example. Surprisingly, we found that the $l_2$ distances to the clean samples were increased after the diffusion models. Not only that we see no signs of shrinkage of distances towards the clean samples, the distances were increase by more than 100%.

This increase in distances as reported above is highly robust. First, it is insensitive to the particular methods used to perform adversarial attack. We tested four different methods (BPDA, PGD, BPDA-

EOT, PGD-EOT), and the results are essentially the same. Second, the results are also robust to the way how distances are measured. Instead of $l_2$ distance, we also quantified the $l_\infty$ distance, and found similar results (see Appendix C.1). The results also hold for multiple datasets we tested (i.e., CIFAR-10 and ImageNet). Finally, the results are robust to how the diffusion-based purification is implemented. In the original DiffPure implementation, both forward (noising) and reverse (denoising) processes were used. In this case, as expected, the DiffPure framework exhibited a clear two-stage process—first increasing the $\ell_2$ distance by the forward diffusion and further decreasing by the reverse denoising (see Fig. 1c). We removed the forward process by only performing denoising, and still observed that the distances to clean samples were increases (not shown).

Thus, diffusion models tend to push the adversarial images further away, rather than being closer to the clean samples as often believed (Nie et al., 2022). As a remark, despite the considerable robustness differences (see Table 1), the $\ell_2$ distances to the clean images were almost identical for different attacks throughout the purification processes. Thus the $\ell_2$ distances to clean samples after diffusion models could not effectively explain the robustness differences. These observations may have important implications for adversarial attacks under diffusion models. It raises the possibility that the attacks based on BPDA may substantially over-estimate the adversarial robustness of the system based on diffusion purification.

**Behavior of diffusion models under random perturbations.** We wonder if the behavior of adversarial attacks under diffusion model is special at all– that is, whether the push-away phenomena we observed is in fact general to arbitrary perturbations around the clean images. To test this, we generated perturbations of clean images by sampling random noise uniformly with a fixed magnitude. We first tested small perturbations that match the size of the adversarial attack on CIFAR-10 ($\ell_\infty = 8/255$ uniform noise). We found that the behavior of the model under random noise (Fig. 1d, blue curve) is almost identical to that induced by the adversarial attack. These results, together with those reported above, suggest that diffusion models are able to reduce the distances to a clean image from a slightly perturbed clean image. This raised the intriguing possibility that the cleans images do not reside on the local peaks of the image priors learned in the diffusion models. This may make sense given the in memorization v.s. generalization trade-off (Kadkhodaie et al., 2024). That is, a model simply encodes every clean image as the prior mode may not generalize well.

Although the results above indicate that diffusion models are ineffective in removing small perturbations, it is possible that they may be more effective in removing noise induced by larger perturbations. We performed the same $\ell_2$ distance analysis using three larger levels of uniform noises, ranging from $\epsilon = \{32, 64, 128\}/255$, to examine the model behavior under larger perturbations (Fig. 1d). As the noise level increases, the $\ell_p$ distances of the final purified states increase. Interestingly, the model transits from "pushing-away" to "shrinkage" under very large perturbations. We repeated the above experiments with $\ell_\infty$ distances (Fig. S1) and on the ImageNet dataset, and confirmed that the observations still held under $\ell_\infty$ measurements across datasets. The original data are provided in Appendix C.1. Overall, these results suggest that, while diffusion models can denoise the image toward the image manifold when the noise is large, locally it is ineffective in removing small noise.

### 3.2 RANDOMNESS LARGELY DETERMINES THE PURIFIED STATES OF DIFFUSION MODELS

Diffusion models are intrinsically noisy due to the noise added in both forward and reverse processes. Thus, a diffusion model defines a stochastic mapping $f_\epsilon$ between the input and output images, i.e., $\mathbf{y} = f_\epsilon(\mathbf{x})$. When the noise $\epsilon$ is fixed (in the implementation, this amounts to fixing the random seeds, Appendix B), the mapping becomes deterministic. Another way to remove the stochasticity in diffusion models is to marginalize over the noise $\epsilon$ by taking the expectation, i.e., $\mathbf{y} = E_{p(\epsilon)}[f_\epsilon(\mathbf{x})]$. Practically, this requires taking a large number of samples and averaging, which is computational expensive. Often, diffusion-based purification is only based on one sample from diffusion models. Thus, we will focus on study the variability of individual samples generated by diffusion models.

We are interested in understanding how randomness in diffusion models determines the states after purification. We consider two relevant scenarios. First, we fix the input image $x$, and pass it through diffusion models multiple times (n = 100) while allowing the randomness to vary from trial to trial. Second, we fix the randomness in diffusion models, and randomly draw input images uniformly from a small ball $\mathbf{M}$ ($\ell_\infty = 8/255$, matching the scale of adversarial attacks) around the clean image $x_0$. As shown in Fig. 2b, starting from the same image, randomness will drive the purified states into

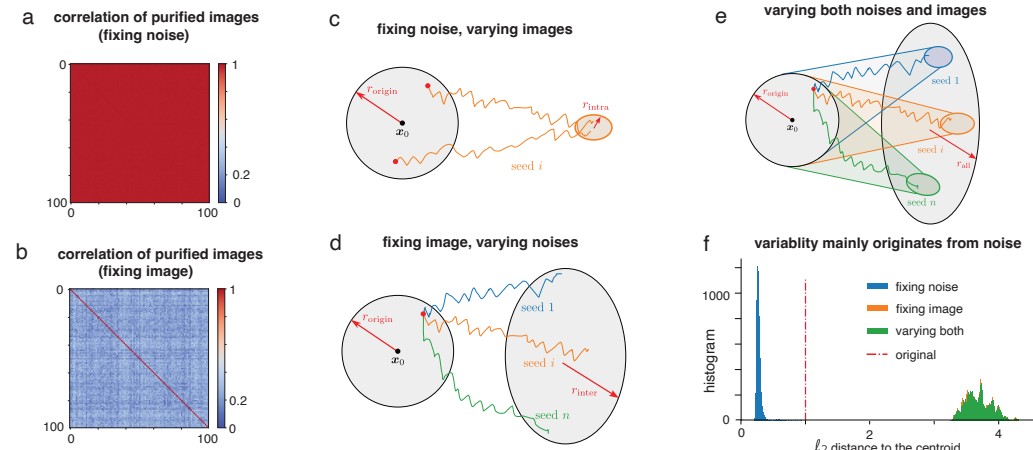

Figure 2: **Randomness largely determines the purified states of diffusion models.** (a) When the noise is fixed, purified images based on images from the image neighborhood are highly correlated. (b) When the input image is fixed, purified images from different samples generated by diffusion models are substantially less correlated. (c,d,e) Schematics illustrating the transformation induced by diffusion models when allowing images or noises to very, or allowing both to vary. (f) The measured distribution of $l_2$ distance of the transformed images to the centroid for three different conditions, comparing to the original perturbations. When noise is fixed, diffusion models reduce the distances. When noise is allowed to vary, diffusion models increase distances.

different (but positively correlated, mean 0.2368) directions. However, even starting from different images within a small ball, the same randomness will drive the purified states into almost exactly the same directions (correlation 0.9954; Fig. 2a). These results imply that if the magnitude of the image perturbation is small (as for the scenario of adversarial examples), randomness is a decisive factor for the eventual purified state. If the randomness is not properly controlled, the gradients calculated for the previous randomness will be applied to an alternative purified state and become non-optimal.

## 4 FUZZY ROBUSTNESS EVALUATION FOR ADVERSARIAL PURIFICATION

### 4.1 FUZZY ADVERSARIAL ROBUSTNESS FOR SYSTEMS WITH RANDOMNESS

The results above suggest that the randomness in diffusion models largely determines the eventual purified state. This implies that randomness may play an important role in the robustness of diffusion models. How randomness affects robustness has been debated. Some relied on randomness as a feature for robustness improvements (Yoon et al., 2021), while others argued it might obscure gradients and make evaluation challenging (Athalye et al., 2018; Carlini et al., 2019; Gao et al., 2022). Here we approach this question from a new perspective. Specifically, we will show that the classical definition for adversarial robustness fall short in evaluating systems with robustness, and it is more appropriate to define "fuzzy adversarial examples". This concept is best illustrated with the following example (also see Fig. 3a).

**Failure of classical adversarial robustness for systems with randomness.** Let $(\boldsymbol{x}_0, y_0)$ be a data point $\boldsymbol{x}_0$ with label $y_0$, and $g(\boldsymbol{x})$ be a deterministic classifier. Consider an additive Gaussian noise model

$$f(\boldsymbol{x}) = \boldsymbol{x} + \boldsymbol{\eta}, \text{where } \boldsymbol{\eta} \sim \mathcal{N}(\boldsymbol{0}, \sigma^2 \boldsymbol{I}). \tag{3}$$

Suppose that we would like to understand the robustness of the system with randomness $s = g \circ f$ around $\boldsymbol{x}_0$. Assume $g$ can successfully classify $\boldsymbol{x}_0$, thus $g(\boldsymbol{x}_0) = y_0$, and $g$ is not a constant, thus $\exists \boldsymbol{x}'$ s.t. $g(\boldsymbol{x}') \neq y_0$. Typically, if $\boldsymbol{x}'$ is close to $\boldsymbol{x}_0$, thus the $\ell_p$-norm less than a given threshold, $\|\boldsymbol{x}' - \boldsymbol{x}_0\|_p < \epsilon$, we say $\boldsymbol{x}'$ is an adversarial example of $g$ around $\boldsymbol{x}_0$. Such a definition works well for the deterministic $g$, but will fail for a system with randomness as $s$. Examining the purification $f(\boldsymbol{x})$, there will be a chance that the clean sample becomes adversarial after purification, $f(\boldsymbol{x}_0) = x'$, or reversely, it turns an adversarial examples back clean, $f(\boldsymbol{x}') = \boldsymbol{x}_0$. The randomness blurs the boundary between adversarial and non-adversarial examples, therefore we can not definitely say that $\boldsymbol{x}'$ is or is not an adversarial example for $s$ anymore.

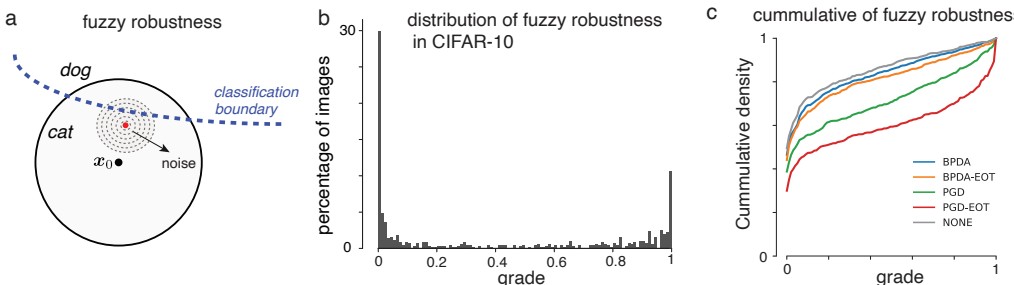

Figure 3: **Fuzzy robustness**. (a) Noise added in classification will make the decision probabilistic and "fuzzy". (b) Distribution of grade from PGD-EOT attack on CIFAR-10 dataset. (c) The cumulative of the fuzzy robustness across the dataset for 4 different attack methods and no attack.

These considerations suggest that it is important to consider the probability of an adversarial attack fooling the system. Below we formalize this idea drawing from the study of fuzzy sets (Zadeh, 1965).

**Definition 1** (Fuzzy adversarial examples). *Let $\{(\boldsymbol{x}_i, y_i)\}$ be a dataset with data $\boldsymbol{x}_i$ and labels $y_i$, $i = 1 \ldots n$. Consider a classification system with randomness $s(\boldsymbol{x})$. For a given threshold $\epsilon$ under $\ell_p$-norm, a perturbed data point $\boldsymbol{\xi}_i$, s.t. $\|\boldsymbol{\xi}_i - \boldsymbol{x}_i\|_p \leq \epsilon$, is said to be a fuzzy adversarial example, with grade*

$$m(\boldsymbol{\xi}_i) = P(s(\boldsymbol{\xi}_i) \neq y_i),$$

*thus, the probability it successfully fools the system. All such fuzzy adversarial examples form a fuzzy set, $\Xi = (U, m)$, where $U = \{\boldsymbol{\xi}_i\}$ is the collection of such perturbed examples and $m(\cdot)$ is the measure function defined above.*

**Definition 2** ($\alpha$-fuzzy adversarial examples). *The fuzzy adversarial example $\boldsymbol{\xi}_i$ is said to be an $\alpha$-fuzzy adversarial example, if*

$$m(\boldsymbol{\xi}_i) \geq \alpha,$$

*thus, with at least probability $\alpha$ it can fool the system. All such $\alpha$-fuzzy adversarial examples form an $\alpha$-cut of the fuzzy set, $\Xi_\alpha = \{\boldsymbol{\xi}_i \in U | m(\boldsymbol{\xi}_i) \geq \alpha\}$.*

### 4.2 FUZZY ROBUSTNESS EVALUATION OF DIFFUSION MODELS

We next evaluate the fuzziness of adversarial examples experimentally in diffusion models. We evaluated the fuzzy robustness for DiffPure on CIFAR-10 and ImageNet dataset. We first calculated the standard BPDA/EOT and PGD/EOT with full gradients on CIFAR-10 ($\ell_\infty = 8/255$), and BPDA/EOT attack on ImageNet ($\ell_\infty = 4/255$) (see Appendix B for details). After calculating the attacks, we repeatedly evaluate the same attack to DiffPure for $r = 100$ times, and use the frequency to estimate the probability of a particular attack fooling the system (i.e., grades of fuzzy adversarial attack). Two extreme cases are of particular interest: (i) the portion of adversarial examples never fool the system (non-adversarial, $m = 0$); (ii) the portion always fools the system successfully (full-adversarial, $m = 1$). If the classical definition still holds approximately, we should expect that the portion of non-adversarial is close to the standard robustness, and full-adversarial is close to the failure rate ($1 -$ standard robustness). The results are shown in Table 1 and 2. The histograms of fuzzy adversarial examples with all grades are shown in Fig. 3b and S4a.

As shown in the Tables and Fig. 3b, rather than being a 0-1 distribution approximating the deterministic regime, the grades have a wide span of range across. Depending on the criteria for considering an image as adversarial (the selection minimum grade $\alpha$), we will have different robustness evaluations rather than a single number believed in the previous works (Nie et al., 2022; Xiao et al., 2023; Lee & Kim, 2023). Since the selection of $\alpha$ is arbitrary, here we set $\alpha = 0.5$ as an exemplar, thus considering an example as adversarial if with at least $50\%$ of the chance it can fool the system. This is a natural selection as for any sample with a grade $m < 0.5$, there is a high chance that it can be corrected by a majority voting (which chooses the category with most "votes" from the samples) .[1]

---

[1]Note that it is not a direct correspondence between samples with grades $m < 0.5$ and that can be corrected by majority voting, unless binary classification, as there can be multiple wrong classes account for the fuzziness.

Table 1: Fuzzy robustness evaluation of DiffPure on CIFAR-10 ($\ell_\infty = 8/255$).

| Attack | Standard Robust. | Fuzzy Robust. | | | Major. Vote | |
|--------|------------------|---------------|--------|---------------|-------------|-----------|
| | | $m = 0$ | $m = 1$ | $\alpha = 0.5$ | $k = 10$ | $k = 100$ |
| Clean | $85.75 \pm 0.82$ | 49.5 | 0.1 | 88.0 | 89.61 | 90.8 |
| BPDA | $83.71 \pm 0.82$ | 46.5 | 0.2 | 85.9 | 87.78 | 88.4 |
| BPDA-EOT | $81.54 \pm 0.84$ | 44.1 | 0.9 | 83.2 | 85.10 | 85.5 |
| PGD (Full) | $71.74 \pm 0.88$ | 38.8 | 1.8 | 70.9 | 74.14 | 74.5 |
| PGD-EOT | $60.53 \pm 0.83$ | 30.0 | 7.0 | 59.4 | 61.58 | 61.2 |

Table 2: Fuzzy robustness evaluation of DiffPure on ImageNet ($\ell_\infty = 4/255$).

| Attack | Standard Robust. | Fuzzy Robust. | | | Major. Vote | |
|--------|------------------|---------------|--------|---------------|-------------|-----------|
| | | $m = 0$ | $m = 1$ | $\alpha = 0.5$ | $k = 10$ | $k = 100$ |
| Clean | $68.70 \pm 1.89$ | 40.0 | 9.5 | 68.0 | 71.0 | 71.5 |
| BPDA | $64.73 \pm 1.79$ | 35.5 | 10.5 | 66.5 | 68.0 | 68.0 |
| BPDA-EOT | $58.41 \pm 1.88$ | 29.0 | 13.5 | 59.0 | 61.5 | 63.0 |

One can define a fuzzy example as adversarial by setting a particular $\alpha$, and acquiring a corresponding robustness. This is essentially the cumulative distribution function of the grade distribution, which we defined as cumulative fuzzy robustness (CFR). Given a particular grade threshold $\alpha$, for the same attack, a higher CFR reflects a better fuzzy robustness; for the same model, a lower CFR reflects a stronger attack. For the limiting case, a perfect robustness system will have an all-zero grade distribution, thus yielding a CFR curve of a horizontal line at 1; while an entirely non-robust system will have an all-one grade distribution, yielding a CFR curve of a horizontal line at 0. Any deterministic system will have a CFR curve with a horizontal line at its robustness rate, while a randomness system will have a CFR curve increasing to 1 (Fig. 3c). We propose the CFR curve as the proper robustness metric for systems with randomness. We observe that the adversarial robustness of the two attacks based on BPDA is substantially lower than that based on PGD. This is consistent with our earlier observation that diffusion models push the adversarial images away from the clean image, which would make the BPDA method less effective.

## 5 THE HYPERSPHERICAL CAP MODEL OF ADVERSARIAL REGIONS

To reveal how adversarial robustness can be improved by purification, it would be important to understand the geometry of the adversarial regions for the classifier. If adversarial regions are tiny regions (e.g., small droplets) and there are many of them surrounding a given clean image, it would be relatively easy to escape these regions by adding a small amount of Gaussian noise. In contrast, if adversarial regions are large and continuous in space, it would be difficult to escape these regions by adding isotropic noise, instead systematic directional biases are needed to avoid these regions. While early work proposed that adversarial regions are small "pockets" in high-dimensional space (Szegedy et al., 2013), Tramèr et al. (2017) showed that rather than being isolated high-dimensional droplets, adversarial examples form continuous spaces so that about 25 orthogonal vectors can be fit into the adversarial region. In the following, we propose a hyperspherical cap model for adversarial regions.

***The hyperspherical cap model.*** An image $x$ is an adversarial example if and only if its projection along the adversarial direction crosses a threshold. Because all such points form a hyperspherical cap around the adversarial direction, the adversarial region is a hyperspherical cap.

Essentially, this model means that for adversarial examples, the classification boundaries are locally linear. To establish this model, we rely on the assumption that adversarial directions are sparse so that they can not be sampled by random directions, which is consistent with prior empirical observations. Under these assumptions, one can mathematically show that the adversarial regions should be a hyperspherical cap when considering $l_2$ neighborhood (Appendix A.1). Notably, based on this model, we can derive an alternative proof of the GAAS algorithm (Tramèr et al., 2017) (Appendix A.2), and as well estimate the volume of adversarial regions for each cap (Appendix A.3).

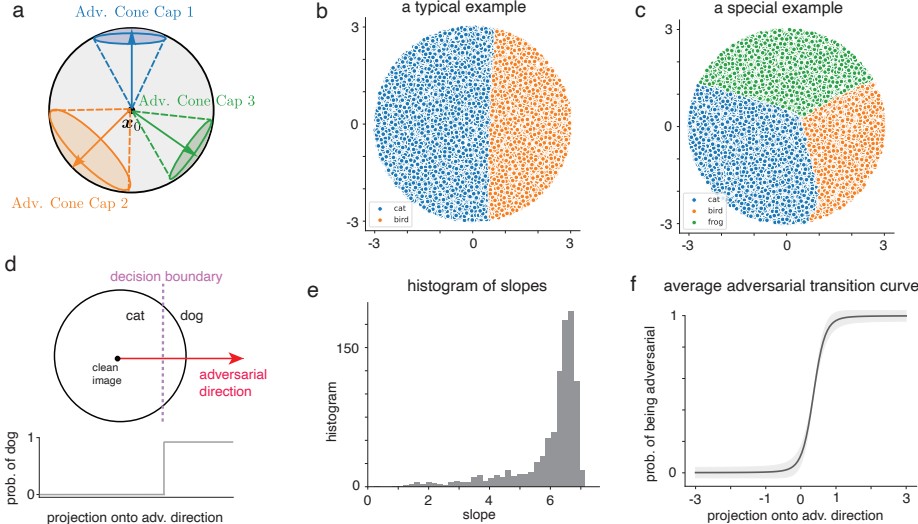

Figure 4: **Hyperspherical cap model.** (a) A schematic of the hyperspherial cap model for adversarial regions. (b) A 2-D projection on one adversarial and one random direction. (c) A 2-D projection on two adversarial directions. (d) A key prediction of the hyperspherical cap model is that the transition of the categorical decision after projecting onto the adversarial direction should be sharp. (e) The empirically measured slope of the transition of decision after projecting onto the adversarial direction. The predominant large slopes support the hyperspherical cap model. (f) The average adversarial transition curve after proper shifting and alignment. This again suggests the transition is sharp.

**Empirical test of the hyperspherical cap model.** We next test the hyperspherical cap model numerically using a WideResNet-28-10 classifier trained on CIFAR-10. We studied its adversarial regions around individual clean images within the $\ell_2$ distances of 3.0. We have deliberately chosen a radius that is larger than the typical length of adversarial attack on CIFAR-10, which is about 1. We first visualized the adversarial region in 2-D subspace spanned by an adversarial direction (via PGD attack, untargeted) and a random direction. Consistent with the hyperspherical cap model, the boundary between the correct and incorrect categories is roughly linear (see Fig. 4b for an example). We also note that, for the special case where slicing over two adversarial directions, two spherical caps may interface with each other (Fig. 4c). We quantify how well the data support the hyperspherical cap model. One key prediction of this model is that, there should be a sharp transition of the categorization decision along the axis of the projection onto adversarial direction (Fig. 4d). We test this prediction by quantifying the sharpness of this transition around 1000 clean images. We found the slope of the transition curve (fit by Sigmoid function) along the adversarial direction is generally large, with most larger than 6 (Fig. 4e). This provides strong empirical support for the hyperspherical cap model and suggests that the decision boundary within the small image neighborhood is roughly linear.

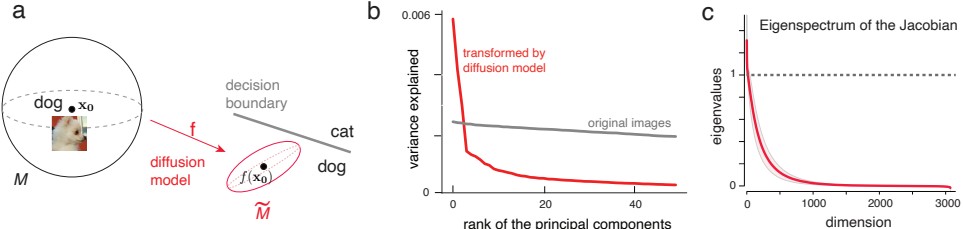

Figure 5: **Diffusion models compress the image space.** (a) Schematic illustrating diffusion model may compress the space to avoid adversarial regions. (b) PCA analysis shows that, after passing through diffusion models, the variance along most direction are reduced. 50 top PCs were plotted. (c) Analysis of the eigenspectrum of the Jacobian at the clean samples again suggest the image neighbourhood is hugely compressed.

## 6 DIFFUSION MODELS IMPROVE ROBUSTNESS BY COMPRESSING IMAGE SPACE

Next, we seek to understand how the diffusion models improve robustness in adversarial purification within a particular randomness configuration. In particular, we would like to understand precisely what factors decide the robust/non-robust outcomes after purification. We have shown in Sec. 3.1 that diffusion models pushes images away from the clean samples, with no shrinkage of distances as naively expected. Intuitively, this may lead to a drop of clean accuracy. Indeed, we observed that clean accuracy drop to about 86% after passing clean images of CIFAR-10 through diffusion models (Table 1). This leads to a conundrum: how can diffusion model increase the robustness after all? We reason that the increase in adversarial accuracy must be due to a narrowing of the gap between the adversarial accuracy and clean accuracy. Crucially, as we will demonstrate below, this narrowing (thus robustness improvement) is due to a compression of image space by diffusion models.

**Diffusion models compress the image neighborhood.** Denote the neighbourhood around a clean sample $x_0$ as $\mathbf{M}$, and the diffusion model transform $\mathbf{M}$ into $\tilde{\mathbf{M}}$ which always contain the transformed clean image $f(\boldsymbol{x}_0)$ (see Fig. 5a). Recall that the clean accuracy is calculated from based on $f(\boldsymbol{x}_0)$, and adversarial accuracy is calculated based on worse-case-scenario in the neighbourhood $\tilde{\mathbf{M}}$. In the extreme case that $\tilde{\mathbf{M}}$ collapses onto $f(\boldsymbol{x}_0)$, the adversarial accuracy and clean accuracy will be identical. This inspired us to treat a purified clean sample $f(\boldsymbol{x}_0)$ as a "reference point", and study the deviation of the worse-case-scenario from it. Specifically, for some perturbation $\boldsymbol{\eta}$, the robustness differences should be decided by the differences of the purified images with respect to the reference points, i.e., $\boldsymbol{\eta}' = f(\boldsymbol{x}_0 + \boldsymbol{\eta}) - f(\boldsymbol{x}_0)$. As hinted in Sec. 3.2, diffusion models may lead to a comparison of the image space, as for random perturbations, the compression rate $\|\boldsymbol{\eta}'\|/\|\boldsymbol{\eta}\| \approx 0.25$. Following this initial observation, we develop two methods to investigate the transformation of image space induced by diffusion models.

*(i) Principal Component Analysis (PCA).* First, we sampled 10,000 points around the neighborhood of a clean image and transformed these images using a diffusion model with fixed randomness. We performed PCA on the original images and the transformed images to obtain their spectrums, respectively. As shown in Fig. 5b, after transformed by the diffusion model, most eigenvalues are much smaller than 1, indicating substantial compression.

*(ii) Analysis of the Jacobian matrix.* Under a locally linear assumption, we may use Taylor expansion to study the transformation induced by diffusion model via interrogating the Jacobian at a clean sample. Specifically, we may write

$$f(\boldsymbol{x}_0 + \boldsymbol{\eta}) \approx \boldsymbol{x}_0 + (f(\boldsymbol{x}_0) - \boldsymbol{x}_0) + J_f(\boldsymbol{x}_0)\boldsymbol{\eta}, \tag{4}$$

Here, the term $f(\boldsymbol{x}_0) - \boldsymbol{x}_0$ describes the shift of the entire image neighborhood as analyzed in Sec. 3.2. Importantly, the eigenspectrum of the Jacobian matrix $J_f(\boldsymbol{x}_0)$ indicates the amount of compression and expansion along each eigen-direction. Fig. 5c plots the averaged eigen-spectrum based on 50 clean images from CIFAR-10. The largest eigenvalue is $1.32 \pm 0.21$, which is comparable with the compression rates; around 90% of the eigenvalues (2760 out of 3072) are smaller than 0.25, with only 17 eigenvalues exceeding 1. Together, the results from the two analyses establish a substantial compression of the image neighborhood induced by diffusion models.

**Dissecting the factors underlying whether an attack would succeed/fail.** What determines whether a particular attack would be successful or not under diffusion models? Given the compression induced by diffusion model, it is natural to consider that the eigenvalues of the Jacobian of the robust images may be smaller that those of the non-robust ones. However, comparing the eigenspectrum of Jacobian matrices for robust v.s. non-robust images, we did not observe a difference between the two.

We next evaluate a couple of additional factors that may distinguish the two (see Fig. 6). First, the closest decision boundary may be closer for some samples compared to others. This can be quantified by moving along the adversarial direction starting from the purified clean sample $f(\boldsymbol{x}_0)$ and identifying the *critical threshold* for reversing the decision to a wrong category. Second, because the orientation of transformed image neighbourhood $\tilde{\mathbf{M}}$ (which can be approximated by an ellipsoid) is likely different for different clean images, it may lead to differences in the *magnitude of the purified adversarial attack*. We find both factors are important.

*(i) Analysis of the critical threshold.* We numerically calculated the critical threshold for robust and non-robust images up to $\ell_2$ distance of 5. We observed that robust samples exhibit higher critical

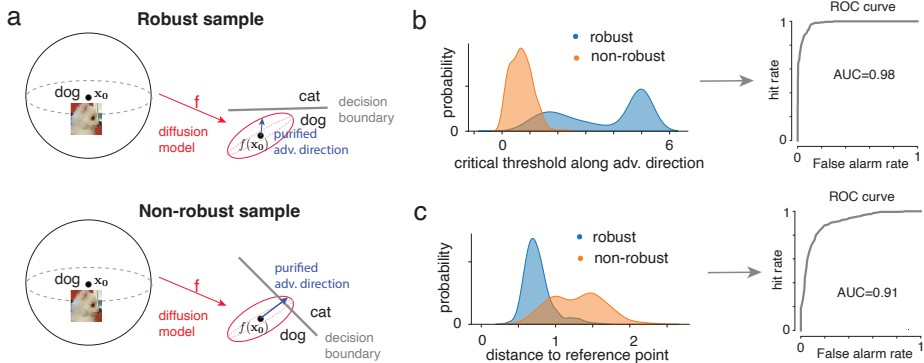

Figure 6: **Robust v.s. non-robust samples.** (a) Schematics showing the factors that make a sample robust or non-robust. The two key factors are the critical threshold along the adversarial direction and the distance of the purified attack to reference point (i.e., the purified clean image). A sample is robust if the former is larger than the latter. (b) Critical threshold along the adversarial direction distinguishes robust v.s. non-robust samples, with a large AUC. (c) Similar to (b), but for distance to the reference point.

thresholds (Fig. 6b). The two distributions are reasonably well separated. We calculated the Receiver operating characteristic (ROC) curve for classifying robustness v.s. non-robustness using the critical threshold alone, and found that the AUC is 0.98, suggesting that the magnitude of the critical threshold is highly predictive of whether a sample is robust or nonrobust.

*(ii) Analysis of the magnitude of the purified attack.* We also observe that there is a significant difference in the length of the purified adversarial perturbations relative to the purified clean samples (Fig. 6c). For robust samples, the average length of purified attack is $0.76 \pm 0.19$. This yield a compression rate of $0.76/1.16 \approx 65.5\%$ compared to the size of the adversarial attack pre-purification. For non-robust samples, the average length of purified attack is $1.29 \pm 0.35$, yielding an expansion rate of $1.29/1.23 \approx 104.9\%$. Analysis of the ROC curve shows that length of the purified adversarial perturbations can distinguish the robust v.s. non-robust samples with an AUC of 0.91, although to a less extent compared to the critical threshold described above.

## 7 DISCUSSION

We have systematically examined the properties of diffusion models used in adversarial purification. We show that diffusion models do not shrinkage the distance of the transformed images toward clean images. In fact, it substantially increases the distance. Furthermore, randomness in diffusion models necessitate a new framework in evaluation of adversarial robustness (e.g., fuzzy robustness). We further show that adversarial regions are large and continuous regions that form hyperspherical caps. Importantly, diffusion models increase robustness by shrinkage of the image space to avoid these regions. Several limitations still exist in our current work. First, we do not offer a principled reason of why diffusion models push the perturbed images further away from clean images. We have speculated that this may be related to the generalization v.s. memorization trade-off in high-dimensional space (Kadkhodaie et al., 2024), which deserves further investigation in the future. Second, we have only tested diffusion models that are formulated in the pixel space. One interesting question is whether these results would generalize to models constructed in latent space (Rombach et al., 2022). Third, our robust evaluation on ImageNet is non-exhausive. Due to the limitation of computational resources, we were unable to run the PGD-EOT attack on ImageNet. Overall, our results clarify a number of misconceptions about adversarial purification and provide new insights into the behavior of diffusion models. Better understanding the exact mechanisms of how diffusion models and diffusion model-based purification work will likely lead to more robust and transparent AI systems.

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

# A THE HYPERSPHERICAL CAP MODEL OF ADVERSARIAL REGIONS

## A.1 THE HYPERSPHERICAL CAP MODEL

Here we describe the hyperspherical cap model of adversarial regions in detail. We start with a formal definition of adversarial directions.

**Definition 3** (Adversarial directions). *Let $(\boldsymbol{x}_0, y_0)$ be the data point-label pair, and consider its robustness in an $\ell_2$ ball with radius $\epsilon$. Let $g(\boldsymbol{x})$ be a classifier correctly classifying the point, i.e. $\arg\max_i g_i(\boldsymbol{x}_0) = y_0$, where $g_i(\boldsymbol{x}_0)$ denotes the i-th logit. For a given direction $\boldsymbol{\eta}$, define the range of the i-th logit (change of the logit) as*

$$r_i(\boldsymbol{\eta}, \epsilon) = \sup_{\gamma_1 \in [0, \epsilon)} g_i(\boldsymbol{x}_0 + \gamma_1 \boldsymbol{\eta}/\|\boldsymbol{\eta}\|_2) - \inf_{\gamma_2 \in [0, \epsilon)} g_i(\boldsymbol{x}_0 + \gamma_2 \boldsymbol{\eta}/\|\boldsymbol{\eta}\|_2).$$

*Then the direction $\boldsymbol{\eta}_{adv}$ is said to be an adversarial direction if for some logit $i$, the change of logit along that direction is considerably larger than the expectation along a random direction $\boldsymbol{\eta}_{rand}$,*

$$\mathbb{E}_{\boldsymbol{\eta}_{rand}} r_i(\boldsymbol{\eta}_{rand}) = o\left(r_i(\boldsymbol{\eta}_{adv})\right).$$

In the following, we propose a hyperspherical cap model for adversarial regions. Essentially, this model relies on the classification boundary being linear locally.

**Assumption 1.** *(Sparsity). Adversarial directions are sparse.*

By "sparse", we mean that adversarial directions usually can not be sampled by random directions. This is in line with the fact that classifiers trained with random Gaussian noises do not provide meaningful improvements on adversarial robustness.

**Assumption 2.** *(Critical threshold). Locally, when moving along the adversarial direction starting from the clean sample, the decision boundary is only crossed once. The radius of the single crossing point (to the clean sample) is defined as the critical threshold $\gamma$.*

**Model 1** (The hyperspherical cap model of adversarial regions.). *When $\epsilon$ is small, consider a point $\boldsymbol{x}$ "around" an adversarial direction $\boldsymbol{\eta}_{adv}$, that is, it has projection onto the adversarial direction*

$$< \boldsymbol{x} - \boldsymbol{x}_0, \boldsymbol{\eta}_{adv}/\|\boldsymbol{\eta}_{adv}\|_2 > = \beta\epsilon = O(\epsilon), \ \beta \in \mathbb{R}. \tag{5}$$

*Based on Assumption 1, the logits at point $\boldsymbol{x}$ can be approximated by the logits of the projection,*

$$g(\boldsymbol{x}) \approx g(\boldsymbol{x}_0 + \beta\epsilon \cdot \boldsymbol{\eta}_{adv}/\|\boldsymbol{\eta}_{adv}\|_2). \tag{6}$$

*Combined with Assumption 2, $\boldsymbol{x}$ is an adversarial example iff the projection crosses the critical threshold, thus $\beta > \gamma/\epsilon$. All such points form a hyperspherical cap around the adversarial direction.*

*Proof.* For the point $\boldsymbol{x}$ around $\boldsymbol{x}_0$, the directional vector $(\boldsymbol{x} - \boldsymbol{x}_0)$ can be decomposite into components parallel and orthogonal to the adversarial direction,

$$(\boldsymbol{x} - \boldsymbol{x}_0) = (\boldsymbol{x} - \boldsymbol{x}_0)_{/\!/\boldsymbol{\eta}_{adv}} + (\boldsymbol{x} - \boldsymbol{x}_0)_{\perp\boldsymbol{\eta}_{adv}},$$

where $(\boldsymbol{x} - \boldsymbol{x}_0)_{/\!/\boldsymbol{\eta}_{adv}} = < \boldsymbol{x} - \boldsymbol{x}_0, \boldsymbol{\eta}_{adv}/\|\boldsymbol{\eta}_{adv}\|_2 > \boldsymbol{\eta}_{adv}/\|\boldsymbol{\eta}_{adv}\|_2 = \beta\epsilon \cdot \boldsymbol{\eta}_{adv}/\|\boldsymbol{\eta}_{adv}\|_2$, and denote

Since $\epsilon$ is small, conduct a Taylor expansion at $\boldsymbol{x}_0 + (\boldsymbol{x} - \boldsymbol{x}_0)_{/\!/\boldsymbol{\eta}_{adv}}$ for some logit $i$,

$$g_i(\boldsymbol{x}) = g_i(\boldsymbol{x}_0 + (\boldsymbol{x} - \boldsymbol{x}_0)_{/\!/\boldsymbol{\eta}_{adv}} + (\boldsymbol{x} - \boldsymbol{x}_0)_{\perp\boldsymbol{\eta}_{adv}})$$
$$\approx g_i(\boldsymbol{x}_0 + (\boldsymbol{x} - \boldsymbol{x}_0)_{/\!/\boldsymbol{\eta}_{adv}}) + \nabla^T g_i(\boldsymbol{x}_0 + (\boldsymbol{x} - \boldsymbol{x}_0)_{/\!/\boldsymbol{\eta}_{adv}}) \cdot (\boldsymbol{x} - \boldsymbol{x}_0)_{\perp\boldsymbol{\eta}_{adv}}.$$

Again since $\epsilon$ is small, assume the gradients do not change considerably locally, then

$$\left|\nabla^T g_i(\boldsymbol{x}_0 + (\boldsymbol{x} - \boldsymbol{x}_0)_{/\!/\boldsymbol{\eta}_{adv}}) \cdot (\boldsymbol{x} - \boldsymbol{x}_0)_{\perp\boldsymbol{\eta}_{adv}}\right| \approx \left|\nabla^T g_i(\boldsymbol{x}_0) \cdot (\boldsymbol{x} - \boldsymbol{x}_0)_{\perp\boldsymbol{\eta}_{adv}}\right| \leq r_i(\perp \boldsymbol{\eta}_{adv}, \epsilon).$$

Since adversarial directions are sparse, with a high probability that $\perp \boldsymbol{\eta}_{adv}$ is not an adversarial direction (Fig. 4b), although we also witness the rare case where $\perp \boldsymbol{\eta}_{adv}$ is indeed another adversarial direction (Fig. 4c), thus

$$\mathbb{E}_{\perp\boldsymbol{\eta}_{adv}} r_i(\perp \boldsymbol{\eta}_{adv}, \epsilon) = o(r_i(\boldsymbol{\eta}_{adv}, \epsilon)).$$

Consequently,

$$g_i(\boldsymbol{x}) \approx g_i(\boldsymbol{x}_0 + \beta\epsilon \cdot \boldsymbol{\eta}_{adv}/\|\boldsymbol{\eta}_{adv}\|_2).$$

$\square$

## A.2 THE GAAS ARRANGEMENT (MAXIMUM $k$-SIMPLEX IN A HYPERSPHERICAL CAP)

Here we provide an alternative proof of the GAAS algorithm (Tramèr et al., 2017) based on the hyperspherical cap model.

*Proof.* For a $(k-1)$ simplex embedded in the $k$ dimensional space with radius $\epsilon$, the coordinates are the permutations of

$$(0, \ldots, \epsilon, \ldots, 0).$$

The center is given by $\left(\frac{\epsilon}{k}, \ldots, \frac{\epsilon}{k}\right)$. Therefore the height (distance to the origin) is

$$h = \frac{\epsilon}{\sqrt{k}}.$$

With the hyperspherical cap model, the height should exceed the critical threshold $\gamma$, denote $\alpha = \gamma/\epsilon$,

$$h = \frac{\epsilon}{\sqrt{k}} \geq \gamma \Rightarrow k \leq \frac{1}{\alpha^2}.$$

Therefore the maximum number of orthogonal directions that can fit into the adversarial regions is

$$k = \min\left(\left\lfloor \frac{1}{\alpha^2} \right\rfloor, d\right),$$

where $d$ is the dimensionality of the data as a trivial bound. $\square$

## A.3 VOLUME OF THE ADVERSARIAL CAP

**Corollary 1.1** (Volume of the adversarial cap). *Let $\bar{\beta} = \gamma/\epsilon$ be the ratio of the critical threshold and the radius of the $\ell_2$ hyperball, and $d$ be the dimensionality of the data. The ratio of volumes between the hyperspherical cap and hyperball is given by (Li, 2010)*

$$\frac{V_{cap}}{V_{ball}} = \frac{1}{2}I_{1-\bar{\beta}^2}\left(\frac{d+1}{2}, \frac{1}{2}\right) = \frac{1}{2}\left(1 - I_{\bar{\beta}^2}\left(\frac{d+1}{2}, \frac{1}{2}\right)\right), \tag{7}$$

*where $I_x(a, b)$ denotes the regularized incomplete beta function (cdf of beta distribution).*

## A.4 FITTING PSYCHOMETRIC FUNCTIONS FOR DECISION BOUNDARIES

We fitted psychometric functions to quantify the transition of decision boundaries in the hyperspherical cap model. For each clean sample, we first picked one adversarial and one random direction to slice a plane from the high-dimensional stimuli space. We further sampled 1000 points within the $\ell_2$ distances of 3 to the clean image on each plane, a typical example is shown in Fig. 4b. Each points are projected onto the adversarial direction (Fig. 4d), and will be measured as either adversarial (1) or non-adversarial (0) to the classifier similar to a two-alternative forced choice (2AFC) task. The binary values and projections onto the adversarial direction were fitted with a sigmoid psychometric function

$$S(x) = \frac{1}{1 + e^{-k(x-x_0)}} \tag{8}$$

with logistic regression, where $k$ is the slope and $x_0$ is the threshold. All fitted psychometric functions (n = 1000, corresponding to the first 10% testing set of CIFAR-10) were aligned to the threshold and averaged, as shown in Fig. 4f.

# B   IMPLEMENTATIONS DETAILS OF ADVERSARIAL ATTACKS ON DIFFPURE

**Datasets and base classifiers**   The experiments were conducted on the CIFAR-10 Krizhevsky & Hinton (2009) and ImageNet (Deng et al., 2009) datasets. For CIFAR-10, we subsampled the first 1000 images from the test set. For ImageNet, we subsampled the first 200 images from the validation set. Standard preprocessing was applied to the datasets. We used the standard classifiers from the RobustBench (Croce et al., 2020)`https://github.com/RobustBench/robustbench`. Namely, the WideResNet-28-10 model for CIFAR-10, and ResNet-50 model for ImageNet. The clean accuracy for the classifier is 94.78% on CIFAR-10 and 76.52% on ImageNet.

**Diffusion models**   We focused on discrete-time diffusion models in this paper to avoid the potential gradient masking induced by numerical solvers in continuous-time models (Huang et al., 2022). For CIFAR-10, we used the official checkpoint of DDPM (coverted to PyTorch from Tensorflow `https://github.com/pesser/pytorch_diffusion`) instead of Score-SDE. For ImageNet, we used the official checkpoint of $256 \times 256$ unconditional Guided diffusion (Dhariwal & Nichol, 2021) `https://github.com/openai/guided-diffusion` as the purification system. The purification time steps were kept the same with Nie et al. (2022), namely $t^* = 0.1$ (100 forward and 100 reverse steps) for CIFAR-10 and $t^* = 0.15$ (150 forward and 150 reverse steps) for ImageNet.

**Fixing randomness in diffusion models**   We controlled the randomness within diffusion models by controlling the random seeds during both the forward and reverse processes. For the base seed $s$, $i$-th batch of data at the $t$ step of the forward/reverse process, we set the random seed

$$\texttt{Seed}(s, i, t) = \begin{cases} 10^4 s + 2 \times (10^3 i + t) + 1, & \text{if forward process} \\ 10^4 s + 2 \times (10^3 i + t), & \text{if reverse process} \end{cases} \tag{9}$$

before sampling the Gaussian noise from eq. 1 or eq. 2. This setting ensures that we have a different random seed for each batch of data and timesteps in the forward/reverse process, but will keep the randomness the same through the entire purification process if encountering the same data batch. We changed the base seed $s$ from 0 to 99 for the inter random seed experiments, and used the base seed 0 for the intra random seed experiments.

**Adversarial attacks**   We conducted BPDA/BPDA-EOT and PGD/PGD-EOT attacks (Athalye et al., 2018) on CIFAR-10 with $\ell_\infty = 8/255$, and BPDA/BPDA-EOT attacks on ImageNet with $\ell_\infty = 4/255$. The PGD was conducted based on the `foolbox` (Rauber et al., 2020)`https://github.com/bethgelab/foolbox`, and the BPDA wrapper was adapted from `advertorch` (Ding et al., 2019)`https://github.com/BorealisAI/advertorch`. Full gradients were calculated for the PGD/PGD-EOT as Lee & Kim (2023) discovered that the approximations methods used in the original DiffPure (Nie et al., 2022) incurred weaker attacks. The full gradient of PGD/PGD-EOT is the strongest attack for DiffPure methods according to Lee & Kim (2023) experiments, and is very computationally expensive. We ran our CIFAR-10 attack experiments on a NVIDIA RTX 6000 GPU for 10 days. We were not able to conduct the full PGD attack on ImageNet in a reasonable time given our available resources. The PGD/PGD-EOT attacks with be made available to the public to facilitate future research. The key hyperparameters for our attacks are listed in Table S1.

Table S1: Hyperparameters for adversarial attacks.

| Hyperparameters | Values {CIFAR-10, ImageNet} |
|---|---|
| Attack magnitude | {8, 4}/255 |
| PGD steps | 40 |
| Relative PGD step size | 0.01 / 0.3 |
| EOT numbers | 15 |
| Batch size | {50, 10} |

# C  ADDITIONAL EXPERIMENTAL RESULTS

## C.1  $\ell_p$ DISTANCE MEASUREMENTS DURING DIFFPURE

Additional distance measurements during the DiffPure process are shown in Fig. S1 and S2. For CIFAR-10, we further measured the $\ell_\infty$ distances for the experiment illustrated in Fig. 1. For ImageNet, we repeated the same experiment with the unconditional Guided diffusion with 150 diffusion and denoising steps ($t^* = 0.15$, the same setting with the DiffPure (Nie et al., 2022)), and measured the $\ell_2/\ell_\infty$ distances. The distances during the intermediate diffusion process in ImageNet (Fig. S2) are not shown as the code base implemented the one-step diffusion equation equivalent to the multistep diffusion. Again, similar effects were observed under both $\ell_2/\ell_\infty$ distances across datasets, namely, diffusion models purified to states further away from the clean images, considerably larger than the original adversarial perturbation ball. Detailed data points are listed in Table S2,S3,S4. Specifically, the $\ell_2/\ell_\infty$ distances to clean samples at the init point ($t = 0$, the scale of the original perturbation), maximum point ($t = 100/150$, after forward diffusion), and end point ($t = 200/300$, after the reverse denoising). The end point distances are roughly 4 or 5 times of the size of the adversarial ball under $\ell_2$ distance on CIFAR-10/ImageNet, and 10 or 26 times under $\ell_\infty$ distance. Diffusion models transit back to the $\ell_2$ shrinkage regime beyond the uniform noise of $\epsilon = 16/255$, which is twice of the standard $\ell_\infty$ adversarial ball considered for CIFAR-10.

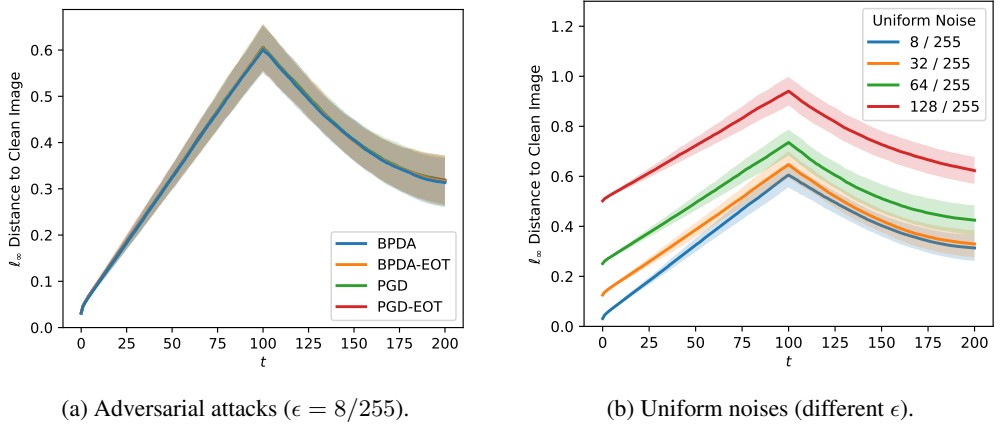

(a) Adversarial attacks ($\epsilon = 8/255$).    (b) Uniform noises (different $\epsilon$).

Figure S1: Diffusion models increase $\ell_\infty$ distances in adversarial purification.

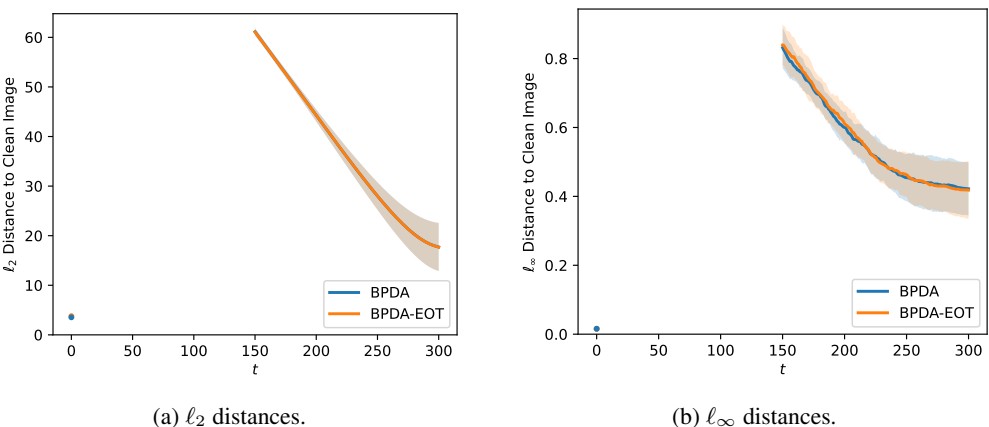

(a) $\ell_2$ distances.    (b) $\ell_\infty$ distances.

Figure S2: Distances measurements during DiffPure on ImageNet.

Table S2: $\ell_2$ distance measurements during DiffPure on CIFAR-10 ($\ell_\infty = 8/255$).

| Attack | Init ($t = 0$) | Max ($t = 100$) | End ($t = 200$) |
|---|---|---|---|
| BPDA | $1.027 \pm 0.023$ | $8.976 \pm 0.118$ | $3.606 \pm 0.607$ |
| BPDA-EOT | $1.072 \pm 0.046$ | $8.992 \pm 0.116$ | $3.607 \pm 0.615$ |
| PGD (Full) | $1.077 \pm 0.040$ | $8.980 \pm 0.118$ | $3.646 \pm 0.614$ |
| PGD-EOT | $1.188 \pm 0.072$ | $8.999 \pm 0.115$ | $3.695 \pm 0.617$ |
| Uniform ($\epsilon = 8/255$) | $1.004 \pm 0.009$ | $8.979 \pm 0.113$ | $3.598 \pm 0.618$ |
| Uniform ($\epsilon = 16/255$) | $4.015 \pm 0.034$ | $9.699 \pm 0.124$ | $3.823 \pm 0.640$ |
| Uniform ($\epsilon = 32/255$) | $8.030 \pm 0.065$ | $11.715 \pm 0.145$ | $5.258 \pm 0.714$ |
| Uniform ($\epsilon = 128/255$) | $16.051 \pm 0.129$ | $17.622 \pm 0.184$ | $9.307 \pm 0.581$ |

Table S3: $\ell_\infty$ distance measurements during DiffPure on CIFAR-10 ($\ell_\infty = 8/255$).

| Attack | Init ($t = 0$) | Max ($t = 100$) | End ($t = 200$) |
|---|---|---|---|
| BPDA | $0.031 \pm 0.000$ | $0.601 \pm 0.051$ | $0.313 \pm 0.053$ |
| BPDA-EOT | $0.031 \pm 0.000$ | $0.603 \pm 0.051$ | $0.316 \pm 0.053$ |
| PGD (Full) | $0.031 \pm 0.000$ | $0.606 \pm 0.050$ | $0.317 \pm 0.055$ |
| PGD-EOT | $0.031 \pm 0.000$ | $0.606 \pm 0.050$ | $0.319 \pm 0.053$ |
| Uniform ($\epsilon = 8/255$) | $0.031 \pm 0.000$ | $0.605 \pm 0.050$ | $0.314 \pm 0.052$ |
| Uniform ($\epsilon = 16/255$) | $0.125 \pm 0.000$ | $0.647 \pm 0.054$ | $0.330 \pm 0.055$ |
| Uniform ($\epsilon = 32/255$) | $0.251 \pm 0.000$ | $0.735 \pm 0.052$ | $0.425 \pm 0.059$ |
| Uniform ($\epsilon = 128/255$) | $0.502 \pm 0.000$ | $0.941 \pm 0.057$ | $0.623 \pm 0.054$ |

Table S4: $\ell_2/\ell_\infty$ distances measurements during DiffPure on ImageNet ($\ell_\infty = 4/255$).

| Distances | Attack | Init ($t = 0$) | Max ($t = 150$) | End ($t = 300$) |
|---|---|---|---|---|
| $\ell_2$ | BPDA | $3.537 \pm 0.079$ | $61.116 \pm 0.738$ | $17.712 \pm 4.851$ |
| | BPDA-EOT | $3.772 \pm 0.139$ | $61.078 \pm 0.762$ | $17.694 \pm 4.838$ |
| $\ell_\infty$ | BPDA | $0.016 \pm 0.000$ | $0.832 \pm 0.059$ | $0.422 \pm 0.077$ |
| | BPDA-EOT | $0.016 \pm 0.000$ | $0.839 \pm 0.060$ | $0.418 \pm 0.084$ |

## C.2 Perceptual metric and additional sampling techniques

In the following sections, we include additional experimental results suggested during rebuttal, which would be incorpated to the main text in the final version.

We repeated the $\ell_p$ distance measurements in Sec. 3.1 with additional sampling techniques along with a perceptual distances. The results are summarized in Table S5. Examples of purified states with diffrent sampling methods are shown in Fig. S3.

Table S5: Distance measurements pre/post-diffusion models on CIFAR-10 (PGD-EOT, $\ell_\infty = 8/255$).

| Sampling method | $\ell_2$ | $\ell_\infty$ | SSIM |
|---|---|---|---|
| DDPG Ho et al. (2020) | $1.188 \to 3.695$ ($\uparrow$) | $0.031 \to 0.319$ ($\uparrow$) | $0.963 \to 0.791$ ($\downarrow$) |
| Reverse-only | $1.188 \to 3.084$ ($\uparrow$) | $0.031 \to 0.273$ ($\uparrow$) | $0.963 \to 0.834$ ($\downarrow$) |
| One-step Carlini et al. (2022) | $1.188 \to 2.692$ ($\uparrow$) | $0.031 \to 0.239$ ($\uparrow$) | $0.963 \to 0.869$ ($\downarrow$) |
| DDIM Song et al. (2020a) | $1.188 \to 2.895$ ($\uparrow$) | $0.031 \to 0.249$ ($\uparrow$) | $0.963 \to 0.861$ ($\downarrow$) |

**Reverse-only diffusion models**  The DiffPure (Nie et al., 2022) framework proposed to utilize both the forward and reverse processes of diffusion models for adversarial purification. Since the forward process introduces a large amount of randomness, we want to explore whether it's possible to remove the forward process, thus only using the reverse process of diffusion models for adversarial purification (RevPure). A similar reverse-only framework was proposed in DensePure (Xiao et al., 2023), but further equipped with a majority voting mechanism to study the certificated robustness.

**Deterministic sampling**  An alternative way to eliminate the effect of randomness is to use a deterministic reverse process. Notably, deterministic reverse process has also been proposed, *e.g.,* in Denoising Diffusion Implicit Models (DDIM) (Song et al., 2020a) the reverse process

$$\boldsymbol{x}_{t-1} = \sqrt{\bar{\alpha}_{t-1}}\hat{\boldsymbol{x}}_0 + \sqrt{1 - \bar{\alpha}_{t-1}}\boldsymbol{\epsilon}_\theta(\boldsymbol{x}_t, t), \ \hat{\boldsymbol{x}}_0 = \frac{\boldsymbol{x}_t - \sqrt{1 - \bar{\alpha}_t}\boldsymbol{\epsilon}_\theta(\boldsymbol{x}_t, t)}{\sqrt{\bar{\alpha}_t}} \tag{10}$$

is fully deterministic and thus does not introduce randomness.[2]

**One-step denoising**  Lastly, we considered the one-step denoising method proposed in Carlini et al. (2022) for improving certificated robustness. One-step denoising significantly reduced computation time, making adversarial attack evaluations more tractable. While clean accuracy remained comparable, adversarial robustness (PGD-EOT) decreased slightly.

The results show that all sampling methods considered (reverse-only, one-step DDPG, and DDIM) achieve comparable clean accuracy to DDPG on CIFAR-10, with $\ell_2$ and $\ell_\infty$ distances to clean images smaller than DDPG but still greater than the initial adversarial perturbations. These findings are consistent with our general results and highlight the robustness of our observations across different sampling methods. We will include these results in the revised manuscript.

**Perceptual-based distances**  Despite the increasement on $\ell_p$ distances to the clean images after purification, it is possible that the actual perceptual-based distances decrease. To study this question, we evaluated the **structural similarity index measure (SSIM)** Wang et al. (2004), a widely used perceptual metric in computer vision, in addition to $\ell_p$ distances. As shown in Table S5, the results reveal an approximate 20% decrease in perceptual distances (SSIM) after diffusion purification, indicating that purified images become perceptually closer to clean images despite the increase in $\ell_p$ distances. This observation complements our findings and will be included in the final paper.

However, we would like to emphasize the importance of $\ell_p$ distance measurements in the context of adversarial purification. This is because adversarial attacks are inherently constructed based on $\ell_p$ distances. Diffusion models do not merely convert adversarial perturbations into smaller $\ell_p$ distances,

---

[2]We follow the notation of the DDPM paper, thus the form is slightly different from the DDIM paper. The $\bar{\alpha}_t$ in DDPM is corresponding to the $\alpha_t$ in DDIM.

which would simplify the problem; instead, they purify states, making them *perceptually closer* to the original images while potentially increasing $\ell_p$ distances. This highlights a unique aspect of their operation that we believe warrants further exploration.

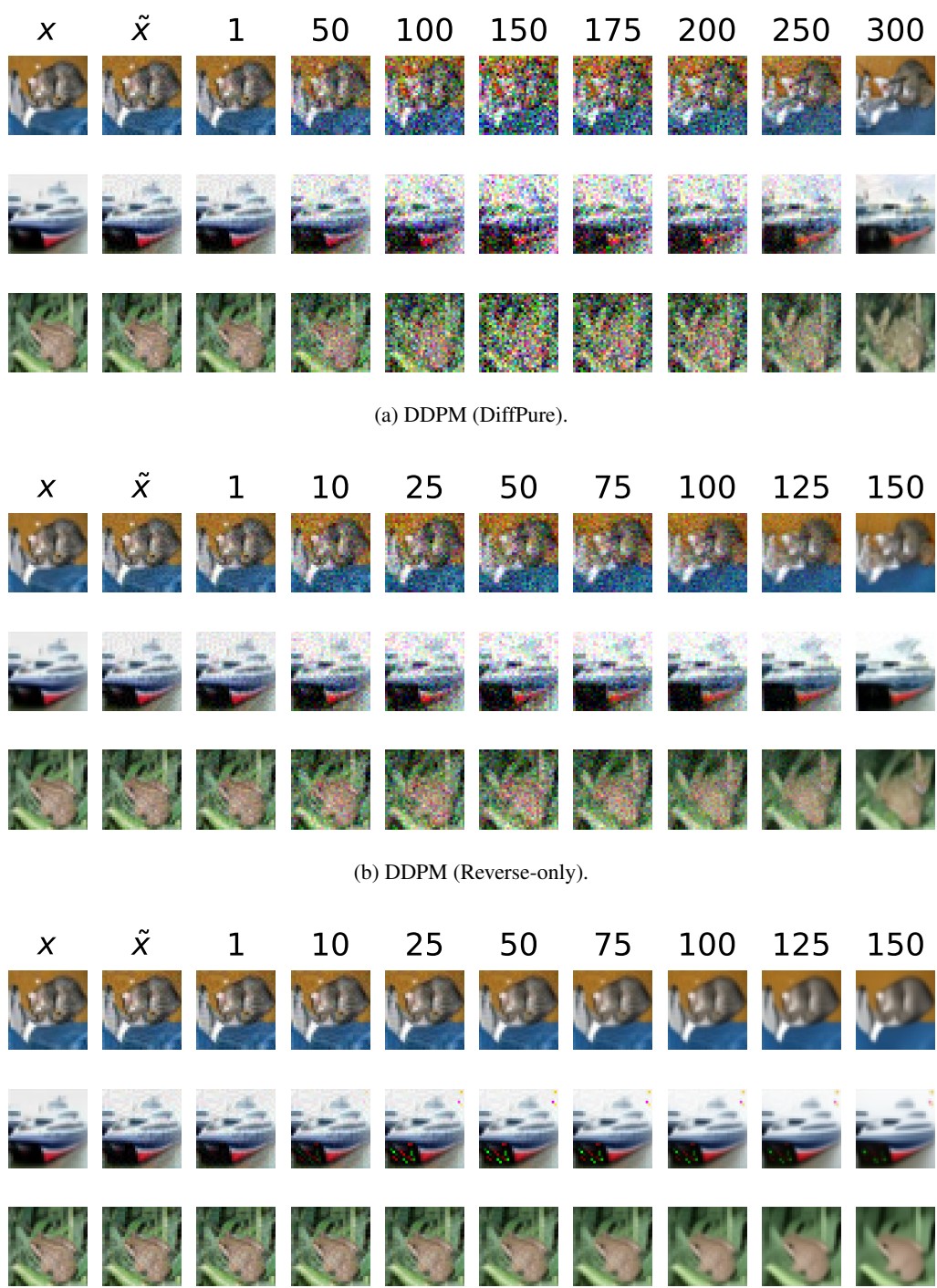

Figure S3: Visualization of the purification process. The first column is the clean stimuli, the second is the perturbed stimuli, and the rest are purified states.

## C.3 FUZZY ROBUSTNESS EVALUATION ON IMAGENET

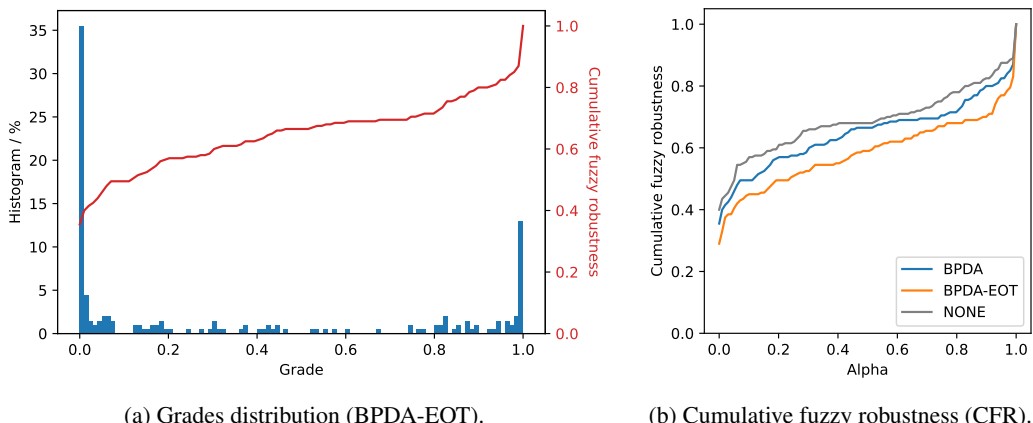

(a) Grades distribution (BPDA-EOT).

(b) Cumulative fuzzy robustness (CFR).

Figure S4: Fuzzy robustness evaluations on ImageNet.

