# OpenReview forum: "How and how well do diffusion models improve adversarial robustness?"
_ICLR.cc/2025/Conference — Submitted to ICLR 2025_

### Official Review · Reviewer_iBpK · 2024-11-01

**Soundness:** 2
**Presentation:** 2
**Contribution:** 3
**Rating:** 5
**Confidence:** 4

**Summary:**

This paper presents several observations and explanations for DiffPure, mainly summarized as follows:

- Diffusion models increase—rather than decrease—the ℓp distances to clean samples.
- The concept of fuzzy adversarial robustness is introduced.
- A hyperspherical cap model of adversarial regions is proposed.
- It is shown that diffusion models increase adversarial robustness by compressing the image space.

**Strengths:**

- By measuring the distance between denoised examples and adversarial images, this paper shows that DiffPure actually increases the distance between denoised images and adversarial examples, rather than decreasing it, which contradicts the previous claim that "DiffPure increases robustness by decreasing the perturbation budget." Although I believe **Carlini et al., Xiao et al., and Nie et al. have never made this claim**, I still consider this experiment insightful, as many people agree with this idea.

- I strongly appreciate the authors' experiments demonstrating that "randomness in diffusion determines the purification direction but not the randomness in the input." I believe this is insightful for diffusion researchers.

**Weaknesses:**

This paper contains several ambiguities, overclaims, and misunderstandings of previous work (see both Weakness and Questions sections), but these can likely be quickly addressed during the rebuttal phase. I would consider raising my score if the authors address these issues.

- I disagree with the authors' claim that "a model simply encoding every clean image as the prior mode may not generalize well" and "the results above indicate that diffusion models are ineffective in removing small perturbations." A large distance between diffusion-purified images and real images does not support this claim. Imagine an image of a panda; the direction of hair growth in pandas can vary, so even if the hair direction changes, the image remains realistic. Such subtle changes often occur after diffusion denoising, causing an increase in ℓp distance, but not due to poor generalization.

- In Fig. 2 (a) and (b), the term "correlation" (e.g., 0.2368, 0.9954 in the paper) is not clearly defined. I'm still unsure of the x and y axes or what is meant by "correlation" here (covariance? cosine similarity? normalized correlation?).

- "Some treated randomness as a bug and proposed methods to cancel its effect in evaluation (cite Carlini et al.)." Carlini never stated this. What Carlini means is that randomness may obscure gradients, making evaluations challenging, but Carlini does not claim that "randomness is a bug" or that randomness inherently negates robustness.

- The authors' definition of "fuzzy adversarial examples" is already extensively discussed in the context of certified robustness via randomized smoothing. You should cite at least [1, 2], as their definition \( g(x)_c = \text{Pr}(f(x) = c) \) closely aligns with yours. While I recognize the differences between your definition and theirs, citing these works is essential for academic integrity.

---

**References**
[1] Cohen, Jeremy, Elan Rosenfeld, and Zico Kolter. "Certified adversarial robustness via randomized smoothing." International Conference on Machine Learning. PMLR, 2019.
[2] Salman, Hadi, et al. "Provably robust deep learning via adversarially trained smoothed classifiers." Advances in Neural Information Processing Systems 32 (2019).

**Questions:**

- I'm unclear about Fig. 1(c). The x-axis is labeled as the diffusion noise level, but what is the y-axis? Is it the distance between noisy examples and the original example, or between denoised examples and the original example? Additionally, why does the image become clearer rather than blurrier as \( t \) increases from 50 to 200? Are these images noisy examples or denoised ones? I can't fully understand this figure, and as a result, I'm unable to accurately interpret lines 162-201.

- "We removed the forward process by only performing denoising and still observed that the distances to clean samples increased." Could you please include this figure? In my opinion, diffusion will produce collapsed images if the input noise level does not align with the diffusion network's noise level condition. I’m really curious to see the result of this experiment.

- "Under these assumptions, one can mathematically show that the adversarial regions should form a hyperspherical cap when considering the \( \ell_2 \) neighborhood." In addition to including the proof in the appendix, I strongly recommend that the authors provide some intuition in the main paper to help readers understand why this holds. Could the authors give further intuition on this result during the rebuttal?

- In Fig. 5(b), does the y-axis represent "variance explained" as the eigenvalue? What is meant by "explained" here?

---

> ### Author Response · Authors · 2024-11-23
>
> We sincerely appreciate your constructive reviews and found them highly insightful. We especially thank you for recognizing the main contributions of our paper, despite the problems in comprehensively interpreting prior works.
>
> ### **Update on the Misclaim**
> Before addressing specific questions, we would like to provide an update regarding a misclaim about the previous explanations of robustness improvement under diffusion models , as this was also a major concern raised by reviewer **aGTV**. Our intent was to convey that prior works have offered intuitive explanations, which *motivated* us to investigate whether diffusion models increase or decrease the $\ell_p$ norms.
>
> By "intuitive explanations", we are referring to statements such as those in the DiffPure framework illustration (Figure 1), where the authors describe diffusion models as “recovering clean images through the reverse denoising process.” We fully acknowledge that prior works did not claim that “diffusion models improve robustness by decreasing $\ell_p$ norms to clean images,” as they did not perform such experiments.
>
> To clarify this point, we plan to rephrase **Sec 3.1** as follows:
>
> > While the exact mechanisms for robustness improvement under diffusion models remain unclear, intuitive explanations have been discussed in the DiffPure paper [Nie et al., 2022], e.g., diffusion models ``recover clean images through the reverse denoising process.'' This motivates us to test a simple hypothesis: diffusion models shrink the $\ell_p$ distances towards clean images during adversarial purification.
>
> We thank both reviewers for pointing out this ambiguity. Please feel free to suggest any further refinements to this section. We believe this rephrasing will adequately resolve the concern and ensure clarity without affecting our experimental results or main arguments.

---

> ### Author Response · Authors · 2024-11-23
>
> ### **Replies to Weaknesses**
>
> * We would like to first clarify that the sentence is not essential to our key results, as it is just a tentative interpretation of our results.  We would be happy to modify or remove it. Here, we are referring to the results from [1] (Figure 2), and the idea is very simple: when the training set is small, diffusion models tend to memorize individual samples in the training set, which is harmful for generalization on the testing set. As the training set increases, diffusion models transition to the generalization phase. The diffusion models typically used are already in this generalization phase, and our experiments further confirmed this observation (not shrinking the $\ell_p$ distances).
>
> In addition, reviewers **7QBf** and **cqB1** suggested measuring perceptual distances in addition to $\ell_p$ distances, and we consider this to be a great idea. We have now measured the **structural similarity index measure (SSIM)** [2] relative to clean images, with the results as follows:
>
> | Distance (to clean images)         | Adversarial (PGD-EOT, $\ell_\infty=8/255$) | Random (uniform, $\ell_\infty=8/255$)       | Purified states        |
> |-------------------------------------|------------------------------------------------------------|-----------------------------------------------------------|----------------------|
> | **SSIM**                            | 0.963 $\pm$ 0.030                                             | 0.966 $\pm$ 0.032                                          | 0.791 $\pm$ 0.085   |
>
> As shown, we observed an approximately 20% decrease under perceptual distances after diffusion purification. This is an interesting complementary result that we will include in the final version of the paper. However, we feel that the  $\ell_p$ distance measurements are still important for understanding adversarial purification, as adversarial attacks are calculated based on $\ell_p$ distances.
>
> * Here we are referring to the Pearson correlation coefficients. Regarding the x-y axes, we sampled either (a) 100 different starting points within the initial adversarial ball using the same random seed during diffusion, or (b) started with the original image but used 100 different random seeds during diffusion. For both settings, we acquired 100 purified states and subtracted the original image to obtain 100 purified vectors. We calculated the Pearson correlation matrix (`np.corrcoef`), resulting in the 100-by-100 matrix shown in the figure.
>
> * We appreciate the reviewer’s deep understanding of Carlini’s work and their clarification, which helped us clarify this issue. We have rephrased the claim “randomness is a bug” as “randomness may obscure gradients, making evaluations challenging” in **Sec. 4.1** accordingly.
>
> * While developing the idea, we also felt it would be beneficial to establish a stronger linkage between randomized smoothing and fuzzy robustness. Despite similarities, there are key differences between these concepts:
>
> 1. **Randomized smoothing** [3] does not necessarily conduct adversarial attacks but relies on numerous evaluations with Gaussian-noisy inputs, categorizing it under *certified robustness*. In contrast, **fuzzy adversarial robustness** evaluates the probability (fuzziness) of adversarial examples for a stochastic system, requiring adversarial attacks first and falling under *empirical adversarial robustness*.
> 2. The **source of randomness** differs: randomized smoothing adds noise to inputs for classifier smoothing, while fuzzy robustness arises inherently from stochastic systems without input noise (no smoothing). Fuzziness is meaningful only for stochastic systems.
> 3. **SmoothAdv** [4], though calculating adversarial attacks, aims to enhance robustness via adversarial training with randomized smoothing, remaining under certified robustness. Fuzzy robustness, however, evaluates robustness directly from the stochastic behavior of the model, without smoothing.
>
> We believe linking the grades of the strongest empirical attack (PGD-EOT) to the certified bound of randomized smoothing would significantly strengthen the concept of fuzzy adversarial robustness. We will add a discussion on this point and add the relevant citations in **Sec. 4.1**.

---

> ### Author Response · Authors · 2024-11-23
>
> ### **Replies to Questions**
>
> * The y-axis in Fig. 1(c) represents the “distances to the original clean image” of purified states using diffusion models. This corresponds to a diffusion model with $t^*=0.1$ in DiffPure on CIFAR-10, involving 100 forward diffusion steps and 100 reverse denoising steps. From steps 0–100, the distances increase due to noise injection; from steps 100–200, the image becomes clearer as the model denoises. The distance at step 0 reflects the initial adversarial perturbation ($\ell_\infty=8/255$, corresponding in $\ell_2$).
>
> * Your intuition is correct: diffusion models perform suboptimally without desired noise levels. This is evident from a drop in clean accuracy when omitting the forward process and only performing the reverse process. This behavior varies by dataset (e.g., a greater drop on ImageNet than CIFAR-10) and checkpoint quality. Below are data from prior experiments:
>
> | Dataset       | Purification Method           | t    | Clean Acc. | BPDA  | BPDA-EOT |
> |---------------|--------------------------------|------|------------|-------|----------|
> | CIFAR-10      | DiffPure (Fix-noise)          | 150  | 79.61      | 68.00 | 67.20    |
> | CIFAR-10      | Reverse-only (Fix-noise)      | 150  | 81.92      | 74.20 | 74.90    |
> | ImageNet      | DiffPure (Fix-noise)          | 150  | 68.50      | 34.00 | --       |
> | ImageNet      | Reverse-only (Fix-noise)      | 100  | 57.10      | 31.00 | --       |
>
> Note: Noise was fixed in these experiments to isolate randomness effects (thus affecting robustness but not clean accuracy).
>
> Overall, reverse-only diffusion achieved better clean accuracy and robustness on CIFAR-10, with a slightly worse robust accuracy on ImageNet. These findings align with our claims, and additional results will be included in the Appendix of the final version.
>
> * Yes, Fig 5(b) shows the explained variance of the PC eigenvectors, corresponding to the PCA analysis. Fig 5(c) displays the eigenvalues of Jacobian matrices, corresponding to the Jacobian analysis. “Explained” refers to the variances of data once projected onto corresponding PC directions.
>
> Thanks again for your review. Please feel free to ask any further questions and we are always happy to discuss.
>
> ***
> ### **References**
>
> [1] Zahra Kadkhodaie, Florentin Guth, Eero P Simoncelli, and Stéphane Mallat. Generalization in diffusion models arises from geometry-adaptive harmonic representations. In The Twelfth International Conference on Learning Representations, 2024.
>
> [2] Wang, Z., Bovik, A. C., Sheikh, H. R., & Simoncelli, E. P. (2004). Image quality assessment: from error visibility to structural similarity. IEEE transactions on image processing, 13(4), 600-612.
>
> [3] Cohen, Jeremy, Elan Rosenfeld, and Zico Kolter. "Certified adversarial robustness via randomized smoothing." International Conference on Machine Learning. PMLR, 2019.
>
> [4] Salman, Hadi, et al. "Provably robust deep learning via adversarially trained smoothed classifiers." Advances in Neural Information Processing Systems 32 (2019).
>
> [5] Florian Tramèr, Nicolas Papernot, Ian Goodfellow, Dan Boneh, and Patrick McDaniel. The space of transferable adversarial examples. arXiv preprint arXiv:1704.03453, 2017.

---

### Official Review · Reviewer_cqB1 · 2024-11-04

**Soundness:** 3
**Presentation:** 2
**Contribution:** 2
**Rating:** 5
**Confidence:** 4

**Summary:**

This paper examined the properties of diffusion models used in defensed against adversarial attacks. The authors showed that diffusion models do not shrinkage the distance of the transformed images toward clean images. Instead, diffusion models force the images to a compressed latent space, thus enhancing adversarial robustness. Altogether, this paper provides an explanation as to why diffusion models improve empirical adversarial robustness.

**Strengths:**

- This paper made an interesting observation on adversarial defenses based on diffusion models, where the purified images have larger l2 distance to the clean images than the adverasrial images. This suggests that diffusion models do not increase robustness by simply removing noise.
- The paper is well-written and motivated, providing clear empirical evidence and concrete discussion as to how diffusion models work on improving adversarial robustness.

**Weaknesses:**

- In Figure 1, the authors show that the l_2 distance to the original clean images increase after diffusion purification. However, the authors do not consider the effect of the sampling process of diffusion models, e.g., deterministic vs random, or more advanced sampling method [1]. It is encouraged to validate the observation on more sampling methods of diffusion models.
- It might be good to have some practical discussions on how the observations/understanding from this paper could contribute to the development of future defenses, or even improving existing diffusion model based defenses.

[1] Elucidating the design space of diffusion-based generative models. 2022.

**Questions:**

- Could the authors evaluate the one-shot denoising approach as used by [1] and validate if the claim that l2 distance between purified images and clean images decrease as compared to that between adversarial images and clean images?
- As indicated by [1], one reason why diffusion model could provided adversarial robustness is that the denoised images can be well classified by the base classifier. Therefore, would it possible that the actual distance decrease under a perceptual based losses [2]?

[1] Certified adversarial robustness for free. Carlini et al, 2023.
[2] Perceptual losses for real-time style transfer and super-resolution. Johnson et al, 2016.

---

> ### Author Response · Authors · 2024-11-27
>
> We thank the reviewer for providing thoughtful comments and constructive feedback. Below, we address your questions and concerns in detail.
>
> ### **Replies to Weaknesses**
>
> * **Advanced Sampling Techniques**
> Sampling techniques beyond DDPG generally involve continuous or deterministic methods.  Continuous methods, such as those solving stochastic differential equations (SDEs) [1], may introduce gradient masking effects during adversarial purification, as observed in adversarial purification with neural ODEs [2]. Hence, our study focuses on discrete sampling methods.  To evaluate deterministic methods, we measured the $\ell_p$ distances of purified images using the DDIM sampling approach [3]:
>
> | Sampling Method | Clean Accuracy | $\ell_2$ to Clean Images | $\ell_\infty$ to Clean Images |
> |-----------------|----------------|--------------------------|-----------------------------|
> | **DDIM**        | 87.9%          | 2.895 ± 0.456            | 0.249 ± 0.041              |
>
> The results show that DDIM achieves comparable clean accuracy to DDPG on CIFAR-10, with $\ell_2$ and $\ell_\infty$ distances to clean images smaller than DDPG but still greater than the initial adversarial perturbations. These findings are consistent with our general results and highlight the robustness of our observations across different sampling methods. We will include these results in the revised manuscript.
>
> * **Future Directions for Diffusion-Based Defenses**
> - The observed **increase in $\ell_p$ distances** (Fig. 1) and the decisive effect of randomness (Fig. 2) suggest that conventional diffusion models, targeted for image generation, operate on variance scales much larger than typical adversarial perturbations. A potential avenue for future work is to train diffusion models tailored to the low-noise regime, transitioning into the **$\ell_p$ shrinkage** regime (Fig. 1d) to establish true attractor dynamics.
>
> - A recent study [4] explored the transition from memorization to generalization in diffusion models, using the bias-free denoising framework [5]. This framework demonstrated that denoising CNNs trained on specific noise levels struggle to generalize to unseen noise levels, an issue mitigated by removing bias terms. Similarly, it is plausible that the lack of $\ell_p$ shrinkage behavior in diffusion models under low noise conditions stems from the presence of bias terms within the U-Net architecture.
>
> - Recent studies on the generalization properties of diffusion models [4], using bias-free denoising frameworks [5], suggest that bias terms in U-Net architectures may contribute to $\ell_p$ distance increases and limit robustness improvements. Investigating these biases systematically may lead to more effective defenses.
>
> - The identified **adversarial compression effect** offers a practical metric for evaluating purification systems without relying on computationally intensive empirical adversarial attacks. This insight could guide the development of more efficient adversarial purification strategies.

---

> ### Author Response · Authors · 2024-11-27
>
> ### **Replies to Questions**
> * **One-Step Denoising**
> We conducted experiments using the one-step denoising approach [6] and obtained the following results:
>
> | Sampling Method | Clean Accuracy | PGD-EOT Robustness | $\ell_2$ to Clean Images | $\ell_\infty$ to Clean Images |
> |------------------|----------------|---------------------|--------------------------|-----------------------------|
> | **DDPG (One-Step)** | 85.2%         | 48.3%               | 2.692 ± 0.444            | 0.239 ± 0.044              |
>
> One-step denoising significantly reduced computation time, making adversarial attack evaluations more tractable. While clean accuracy remained comparable, adversarial robustness (PGD-EOT) decreased slightly, consistent with our findings. These results reinforce the broader applicability of our conclusions.
>
> * **Perceptual-Based Metrics**
> Calculating the perceptual-related distances is a great idea, as also suggested by the reviewer **7QBf** (FID score). To study this question, we evaluated the **structural similarity index measure (SSIM)** [7], a widely used perceptual metric in computer vision, in addition to $\ell_p$ distances. The results are summarized below:
>
> | Distance to Clean Images         | Adversarial (PGD-EOT, $\ell_\infty=8/255$) | Random (Uniform, $\ell_\infty=8/255$) | Purified States      |
> |----------------------------------|-------------------------------------------|---------------------------------------|----------------------|
> | **SSIM**                         | 0.963 ± 0.030                            | 0.966 ± 0.032                        | 0.791 ± 0.085        |
>
> The results reveal an approximate 20% decrease in perceptual distances (SSIM) after diffusion purification, indicating that purified images become perceptually closer to clean images despite the increase in $\ell_p$ distances. This observation complements our findings and will be included in the final paper.
>
> However, we would like to emphasize the importance of $\ell_p$ distance measurements in the context of adversarial purification. This is because adversarial attacks are inherently constructed based on $\ell_p$ distances. Diffusion models do not merely convert adversarial perturbations into smaller $\ell_p$ distances, which would simplify the problem; instead, they purify states, making them **perceptually closer** to the original images while potentially increasing $\ell_p$ distances. This highlights a unique aspect of their operation that we believe warrants further exploration.
>
> ---
> ### **References**
> [1] Song, Y., Sohl-Dickstein, J., Kingma, D. P., Kumar, A., Ermon, S., & Poole, B. (2020). Score-based generative modeling through stochastic differential equations. arXiv preprint arXiv:2011.13456.
>
> [2] Huang, Y., Yu, Y., Zhang, H., Ma, Y., & Yao, Y. (2022, April). Adversarial robustness of stabilized neural ode might be from obfuscated gradients. In Mathematical and Scientific Machine Learning (pp. 497-515). PMLR.
>
> [3] Jiaming Song, Chenlin Meng, and Stefano Ermon. Denoising diffusion implicit models. In International Conference on Learning Representations, 2020a
>
> [4] Zahra Kadkhodaie, Florentin Guth, Eero P Simoncelli, and Stéphane Mallat. Generalization in diffusion models arises from geometry-adaptive harmonic representations. In The Twelfth International Conference on Learning Representations, 2024.
>
> [5] Mohan, S., Kadkhodaie, Z., Simoncelli, E. P., & Fernandez-Granda, C. (2019). Robust and interpretable blind image denoising via bias-free convolutional neural networks. arXiv preprint arXiv:1906.05478.
>
> [6] Carlini, N., Tramer, F., Dvijotham, K. D., Rice, L., Sun, M., & Kolter, J. Z. (2022). (certified!!) Adversarial robustness for free!. arXiv preprint arXiv:2206.10550.
>
> [7] Wang, Z., Bovik, A. C., Sheikh, H. R., & Simoncelli, E. P. (2004). Image quality assessment: from error visibility to structural similarity. IEEE transactions on image processing, 13(4), 600-612.

---

### Official Review · Reviewer_pJXW · 2024-11-04

**Soundness:** 3
**Presentation:** 4
**Contribution:** 4
**Rating:** 8
**Confidence:** 2

**Summary:**

As mentioned in the title, this paper aims to investigate how and how well diffusion models improve adversarial robustness. Specifically, this paper first demonstrates that diffusion models push the purified images away from clean images, which challenges the previous belief (i.e., diffusion models will push adversarial images closer to clean images).  Then, this paper shows that the randomness largely influences the robustness of diffusion models. Based on this, this paper introduces a new concept called 'fuzzy adversarial examples', which considers the probability of adversarial attack fooling the system and proposes a new robust evaluation metric called 'cumulative fuzzy robustness (CFR)' to evaluate the robustness of fuzzy adversarial examples. Lastly, this paper uses a hyperspherical cap model to show that diffusion models improve robustness by shrinking the image space.

**Strengths:**

1. This paper is very well-written and easy to follow.

2. This paper provides sufficient insights (as mentioned in the summary) on diffusion-based adversarial purification, which can inspire researchers in this area to develop more advanced defense methods. More importantly, this paper closes an important research gap on how diffusion-based adversarial purification actually works. I would also like to hear other reviewers' opinions on the contributions of this paper.

3. The concept of fuzzy adversarial examples is novel and intuitive. Based on this, the proposed cumulative fuzzy robustness (CFR) is a more suitable evaluation metric to examine the robustness of a system with inherent robustness (especially for diffusion-based adversarial purifications).

**Weaknesses:**

1. While the observation that the diffusion model further pushes adversarial examples away from clean examples is intriguing, this paper lacks a theoretical explanation of why this occurs. However, this is only a minor weakness, as including a theoretical explanation is not always possible.

2. Fuzzy robustness evaluation is only performed on DiffPure, which is a bit outdated. Authors are encouraged to evaluate more recent diffusion-based adversarial purification methods to make results more convincible (e.g., [1] [2]).

[1] DensePure: Understanding Diffusion Models Towards Adversarial Robustness, ICLR 2023.
[2] Robust Evaluation of Diffusion-Based Adversarial Purification, ICCV 2023.

**Questions:**

1. What is the PGD+EOT result on ImageNet?

2. Just curious—do you think the conclusions in this paper would hold up if diffusion models are constructed in latent space instead of pixel space? And why?

---

> ### Author Response · Authors · 2024-11-24
>
> We appreciate your positive feedback and are glad to see that our work might be insightful for the community. As you mentioned being interested in other reviewers’ opinions, we would like to first summarize our updates based on common questions raised by the reviewers.
>
> ### **Summary of Updates**
>
> * **Clarification on the Misclaim**
> As noted by reviewers **aGTV** and **iBpK**, we clarified the misclaim regarding $\ell_p$ norms in **Sec. 3.1**. Specifically, we rephrased that paragraph:
>   > While the exact mechanisms for robustness improvement under diffusion models remain unclear, intuitive explanations have been discussed in the DiffPure paper [Nie et al., 2022], e.g., diffusion models ``recover clean images through the reverse denoising process.'' This motivates us to test a simple hypothesis: diffusion models shrink the $\ell_p$ distances towards clean images during adversarial purification.
>
> This addresses concerns about ambiguity and better aligns with prior works without impacting experimental results or key arguments.
>
> * **New Perceptual Distance Measurements**
> Responding to feedback from reviewers **7QBf** and **cqB1**, we included **structural similarity index measure (SSIM)** [1] results alongside $\ell_p$ distances. These measurements provide complementary insights into the purification process and will be incorporated into the final version.
>
>   | Distance (to clean images)         | Adversarial (PGD-EOT, $\ell_\infty=8/255$) | Random (uniform, $\ell_\infty=8/255$)       | Purified states        |
>   |-------------------------------------|------------------------------------------------------------|-----------------------------------------------------------|----------------------|
>   | **SSIM**                            | 0.963 $\pm$ 0.030                                             | 0.966 $\pm$ 0.032                                          | 0.791 $\pm$ 0.085   |
>
> * **Additional Results of Reverse-Only Diffusion and DDIM Sampling**
> Responding to feedback from reviewers **7QBf** and **iBpK**, we included additional experimental results with reverse-only diffusion models in the appendix. Results with other sampling techniques, such as **DDIM** [2], were also appended in response to reviewer **cqB1**, as other SDE-based samplings may raise concerns in gradient masking from the numerical solver [3].
>
> * **Discussion on Fuzzy Adversarial Robustness**
> We elaborated on the distinction between **randomized smoothing** [4] and **fuzzy adversarial robustness**, emphasizing differences in their motivation, procedure, and source of randomness. The added discussion strengthens the conceptual framing and includes citations to related works, such as **SmoothAdv** [5] and **DensePure** [6].
>
> The remaining questions are addressed separately in our detailed responses.

---

> ### Author Response · Authors · 2024-11-24
>
> ### **Replies to the Weaknesses**
>
> * Yes, we acknowledge that the theoretical explanation in our work is not entirely comprehensive. However, we believe it represents a step forward toward understanding the role of diffusion models in adversarial purification. Specifically:
>   - We developed the **hyperspherical cap model** of adversarial regions (Sec. 5).
>   - We identified the **adversarial compression effect** (Sec. 6).
>   These insights allowed us to pinpoint two key factors influencing robustness differences across individual images:
>   1. **Adversarial Compression Rate**—dictated by the purification system.
>   2. **Critical Threshold**—determined by the classification system.
>   Together with the decisive effect of randomness (Fig. 2), these aspects were not sufficiently recognized by previous studies.
>
> * Following the suggestions from [7], we implemented PGD-EOT attacks as an approximation of AutoAttack and found them generally effective. Additionally, we want to emphasize the distinctions between **fuzzy robustness** and **DensePure** [6]:
>   - **DensePure** uses the reverse process of diffusion models as an off-the-shelf denoiser within the denoised smoothing framework [8]. It relies on numerous evaluations with Gaussian-noisy inputs and belongs to the category of *certified robustness*.
>   - In contrast, **fuzzy adversarial robustness** evaluates the probability (*fuzziness*) of adversarial examples in stochastic systems. It requires conducting adversarial attacks first and thus falls under *empirical adversarial robustness*.
>   While there are some conceptual connections, applying the notion of fuzzy adversarial robustness to DensePure may not be entirely appropriate.
>
> ### **Replies to the Questions**
>
> * **PGD-EOT attacks on ImageNet**
>   Unfortunately, calculating full gradients on ImageNet with diffusion models would not be feasible given our our computational resources. As noted in Appendix B, it took around 10 days to compute full PGD-EOT gradients on CIFAR-10 with 100 denoising steps. ImageNet images are 64 times larger and require 1.5× times of denoising steps, which would not be pratical with our current setup.
>   This limitation has been highlighted in prior works and underscores the challenge of properly estimating robustness at this scale [7]. As a result, we only included BPDA-EOT results for ImageNet in our submission.
>
> * **Behavior of $\ell_p$ distances in latent space**
>   This is an excellent question. We assume you were referring to latent structures similar to those studied in diffusion models [9]. We speculate that the behavior would likely depend on the relative scale of the latent space:
>   - If the latent space scale is similar to the image space, there may be an **increase** in $\ell_p$ distances.
>   - If the latent space has a larger scale comparable to the inherent variation induced by the diffusion process (as in Fig. 1d), there may be a **decrease** in $\ell_p$ distances.
>   This opens an interesting avenue for future investigation.
>
> Please feel free to ask any further questions and we are always happy to discuss.
> ***
> ### **References**
>
> [1] Wang, Z., Bovik, A. C., Sheikh, H. R., & Simoncelli, E. P. (2004). Image quality assessment: from error visibility to structural similarity. IEEE transactions on image processing, 13(4), 600-612.
>
> [2] Jiaming Song, Chenlin Meng, and Stefano Ermon. Denoising diffusion implicit models. In International Conference on Learning Representations, 2020a
>
> [3] Huang, Y., Yu, Y., Zhang, H., Ma, Y., & Yao, Y. (2022, April). Adversarial robustness of stabilized neural ode might be from obfuscated gradients. In Mathematical and Scientific Machine Learning (pp. 497-515). PMLR.
>
> [4] Cohen, Jeremy, Elan Rosenfeld, and Zico Kolter. "Certified adversarial robustness via randomized smoothing." International Conference on Machine Learning. PMLR, 2019.
>
> [5] Salman, Hadi, et al. "Provably robust deep learning via adversarially trained smoothed classifiers." Advances in Neural Information Processing Systems 32 (2019).
>
> [6] Xiao, C., Chen, Z., Jin, K., Wang, J., Nie, W., Liu, M., ... & Song, D. (2023). Densepure: Understanding diffusion models for adversarial robustness. In The Eleventh International Conference on Learning Representations.
>
> [7] Robust Evaluation of Diffusion-Based Adversarial Purification, ICCV 2023.
>
> [8] Salman, H., Sun, M., Yang, G., Kapoor, A., & Kolter, J. Z. (2020). Denoised smoothing: A provable defense for pretrained classifiers. Advances in Neural Information Processing Systems, 33, 21945-21957.
>
> [9] Chen, X., Liu, Z., Xie, S., & He, K. (2024). Deconstructing denoising diffusion models for self-supervised learning. arXiv preprint arXiv:2401.14404.

---

### Official Review · Reviewer_7QBf · 2024-11-04

**Soundness:** 3
**Presentation:** 3
**Contribution:** 2
**Rating:** 5
**Confidence:** 4

**Summary:**

This is an analytical work to investigate how diffusion-based purification (DBP) improve the adversarial robustness. With pilot experiments, this work finds that diffusion models increase the lp distances to clean samples and the purified images are heavily influenced by the internal randomness of diffusion models. Furthermore, this work introduces the concept of fuzzy adversarial robustness and hyper-spherical cap model of adversarial regions, and gives an explanation on how DBP works.

**Strengths:**

1.	Overall, this paper is well-written, with clear organizations and illustrations.
2.	The finds in this work are reasonable, with sufficient explanations and experimental supports.

**Weaknesses:**

My major concern on this work is about the contribution of this work:

1)	It is not surprising that diffusion-based purification will guide the adversarial images to a place with a larger distance to the original images. As the adversarial perturbation is usually small and the corrupted process (adding Gaussian noise) will make it hard to reverse it to the original image (information on the original image is missing). According to [1,2], the reversion process of diffusion models is to make the generated image has a higher probability to be close to the original image distribution, which does not mean this image should be closer to the original image. Below are some suggested experiments to make this work more comprehensive:

- a)	Besides traditional l_p distance, the metrics for generative models (e.g., FID score, Inception score) should also be used to evaluate the distance. In such latent space, the conclusion might be different.

- b)	Lines 171-172. I think this experiment is important and should not be hidden.

- c)	Early denoiser-based defense [3] has been shown to be ineffective while diffusion-based purification is effective. It is important to show if early denoiser-based defense has the similar behavior (enlarging the lp distance) to diffusion models. If they are similar, the findings in this work may not explain why they have different defense performance.

2)	The proposed fuzzy adversarial robustness (a new framework in evaluation) is similar to [1] while the discussion on it is missing.
3)	This work claims that the findings offer guidance for the development of more efficient adversarial purification (AP) systems, but no deeper discussions and experiments are provided. If this work can give a prototype method of more efficient AP with the findings, I could have improved my score.

Line 450: $\textbf{x}_0$， line 269: $\textbf{x}^{`}$

[1] Xiao C, Chen Z, Jin K, et al. Densepure: Understanding diffusion models towards adversarial robustness[J]. arXiv preprint arXiv:2211.00322, 2022.

[2] Chen H, Dong Y, Wang Z, et al. Robust classification via a single diffusion model[J]. arXiv preprint arXiv:2305.15241, 2023.

[3] Liao F, Liang M, Dong Y, et al. Defense against adversarial attacks using high-level representation guided denoiser[C] CVPR. 2018: 1778-1787.

[4] DiffHammer: Rethinking the Robustness of Diffusion-Based Adversarial Purification. NeurIPS, 2024

**Questions:**

Please see the weaknesses above.

---

> ### Author Response · Authors · 2024-11-29
>
> We thank the reviewer for providing invaluable suggestions on improvements. Below, we address the weaknesses and concerns in detail.
>
> ### **Replies to Weaknesses**
> 1. **Follow-up Experiments on $\ell_p$ Distances Measurements**
> * Perceptual-based metrics
>
> Calculating the perceptual-related distances is a great idea, as also suggested by the reviewer **cqB1** (perceptual loss). To study this question, we evaluated the **structural similarity index measure (SSIM)** [1], a classical perceptual metric in image processing and vision neuroscience, in addition to $\ell_p$ distances. The results are summarized below:
>
> | Distance to Clean Images         | Adversarial (PGD-EOT, $\ell_\infty=8/255$) | Random (Uniform, $\ell_\infty=8/255$) | Purified States      |
> |----------------------------------|-------------------------------------------|---------------------------------------|----------------------|
> | **SSIM**                         | 0.963 ± 0.030                            | 0.966 ± 0.032                        | 0.791 ± 0.085        |
>
> The results reveal an approximate 20% decrease in perceptual distances (SSIM) after diffusion purification, indicating that purified images become perceptually closer to clean images despite the increase in $\ell_p$ distances. This observation complements our findings and will be included in the final paper.
>
> However, we emphasize the importance of $\ell_p$ distance measurements in the context of adversarial purification. This is because adversarial attacks are constructed inherently based on $\ell_p$ distances. Diffusion models do not merely convert adversarial perturbations into smaller $\ell_p$ distances, which would simplify the problem; instead, they purify states, making them **perceptually closer** to the original images while potentially increasing $\ell_p$ distances. This highlights a unique aspect of their operation that we believe warrants further exploration in the future.
>
> * Reverse-only diffusion
>
> Thanks for pointing it out. We have performed these analyses, and add the results of reverse-only diffusion in Appendix C2. A brief summary is provided below:
>
>
> | Sampling Method | $\ell_2$ to Clean Images | $\ell_\infty$ to Clean Images | SSIM |
> |------------------|----------------|---------------------|--------------------------|
> | **DDPG (Reverse-only)** |  1.188 $\to$ 3.084  ($\uparrow$) | 0.031 $\to$ 0.273  ($\uparrow$) | 0.963 $\to$ 0.834 ($\downarrow$) |
>
> As shown, reverse-only diffusion also results in increased $\ell_p$ distances and decreased SSIM after purification. This consistency supports the generalizability of our conclusions.
>
> * Denoiser-based defense
>
> While we have not conducted experiments to compare $\ell_p$ distances in denoiser-based defenses, we clarify that our goal is not to claim that increased $\ell_p$ distances are either (i) unique to diffusion models, or (ii) critical for robustness improvements. Instead, our findings rule out the conventional hypothesis that diffusion models act as $\ell_p$ denoisers to convert adversarial perturbations into smaller norms. We hypothesize that differences in robustness may be attributable to adversarial compression rates (Sec 6), though the inherent randomness of diffusion models poses additional challenges in evaluation (Sec 3.2 and Sec 4).
>
> 2. **Discussion on Fuzzy Robustness**
>
> This point is also suggested by reviewer iBpK. To address this question, we add a discussion with our fuzzy adversarial robustness with randomized smoothing [2] (including DensePure [3]). Despite similarities, there is a key difference between these concepts: **randomized smoothing** does not necessarily conduct adversarial attacks but relies on numerous evaluations with Gaussian-noisy inputs, categorizing it under *certified robustness*. In contrast, **fuzzy adversarial robustness** evaluates the probability (fuzziness) of adversarial examples for a stochastic system, requiring adversarial attacks first and falling under *empirical adversarial robustness*.
>
> We believe linking the grades of the strongest empirical attack (PGD-EOT) to the certified bound of randomized smoothing would significantly strengthen the concept of fuzzy adversarial robustness. We will add a discussion on this point and add the relevant citations in the final version.

---

> ### Author Response · Authors · 2024-11-29
>
> 3. **Future Directions for Diffusion-Based Defenses**
> - The observed **increase in $\ell_p$ distances** (Fig. 1) and the decisive effect of randomness (Fig. 2) suggest that conventional diffusion models, targeted for image generation, operate on variance scales much larger than typical adversarial perturbations. A potential avenue for future work is to train diffusion models tailored to the low-noise regime, transitioning into the **$\ell_p$ shrinkage** regime (Fig. 1d) to establish true attractor dynamics.
>
> - A recent study [4] explored the transition from memorization to generalization in diffusion models, using the **bias-free denoising** framework [5]. It was shown that the bias terms in U-Net architectures hindered the denoising model from generalizing to other unseen noise levels. As we illustrated the scales of adversarial perturbation were considerably larger than the inherent randomness of diffusion models, the bias terms in the diffusion model might limit robustness improvements. Investigating these biases could lead to more effective defenses.
>
> - The identified **adversarial compression effect** offers a practical metric for evaluating purification systems without relying on computationally intensive empirical adversarial attacks. This insight could guide the development of more efficient adversarial purification strategies.
>
> Please feel free to ask further questions and we are happy to receive your feedback.
>
> ---
> ### **References**
> [1] Wang, Z., Bovik, A. C., Sheikh, H. R., & Simoncelli, E. P. (2004). Image quality assessment: from error visibility to structural similarity. IEEE transactions on image processing, 13(4), 600-612.
>
> [2] Cohen, Jeremy, Elan Rosenfeld, and Zico Kolter. "Certified adversarial robustness via randomized smoothing." International Conference on Machine Learning. PMLR, 2019.
>
> [3] Xiao C, Chen Z, Jin K, et al. Densepure: Understanding diffusion models towards adversarial robustness[J]. arXiv preprint arXiv:2211.00322, 2022.
>
> [4] Zahra Kadkhodaie, Florentin Guth, Eero P Simoncelli, and Stéphane Mallat. Generalization in diffusion models arises from geometry-adaptive harmonic representations. In The Twelfth International Conference on Learning Representations, 2024.
>
> [5] Mohan, S., Kadkhodaie, Z., Simoncelli, E. P., & Fernandez-Granda, C. (2019). Robust and interpretable blind image denoising via bias-free convolutional neural networks. arXiv preprint arXiv:1906.05478.

---

> > ### Comment · Reviewer_7QBf · 2024-12-02
> >
> > Thanks for the further experiments. I tend to maintain my current score. It is worth noting that adversarial examples are beyond l_p norm. For example, there are patch-attack and diffusion-based attacks. The different conclusions between lp distance and perceptual-based metrics make me worry about the generalization of the conclusion, especially targeting the attacks beyond l_p attack.

---

> > > ### Author Response · Authors · 2024-12-04
> > >
> > > We sincerely thank the reviewer for their invaluable feedback. Your concern regarding distances beyond $\ell_p$ norms in the context of adversarial robustness is valid and potentially opens an interesting avenue for further exploration. While we acknowledge that $\ell_p$ distances may not encompass all possible measures of robustness, we believe that they address the majority of cases relevant to adversarial attacks. Additionally, our findings showing an increase in $\ell_p$ distances alongside a decrease in perceptual-based metrics (e.g., SSIM) provide a compelling contrast that highlights the unique behavior of current diffusion models. Nonetheless, we recognize this topic is open to debate and welcome further discussion.
> > >
> > > We also would like to emphasize that $\ell_p$ distances form only a small portion of our overall results. They serve as an intriguing starting point for introducing the concept of **reference points** (purified clean images) in Section 6. Beyond this, we feel that the decisive effect of randomness discussed in Section 3.2, the exploration of fuzzy robustness in Section 4, and the adversarial cap model and compression effects detailed in Sections 5 and 6 are equally critical to our contribution.
> > >
> > > We encourage further discussion among the reviewers on these points and would be happy to provide additional clarifications or address any further questions. Thank you once again for your thoughtful feedback.

---

### Official Review · Reviewer_aGTV · 2024-11-04

**Soundness:** 2
**Presentation:** 3
**Contribution:** 2
**Rating:** 5
**Confidence:** 4

**Summary:**

This paper provides theoretical and empirical studies on the mechanism of diffusion-based adversarial purification methods. First, based on empirical studies on the behavior of diffusion-based purification models, it is suggested that the purification generally increases the $\ell_p$ distance of the input sample to the clean sample, and that the purification results are affected by randomness in diffusion models. Second, the concept of fuzzy robustness and the corresponding evaluation method are proposed for diffusion-based purification methods. Third, a hyperspherical cap model is introduced to depict the adversarial regions. Finally, it is argued that the robustness brought by diffusion-based purification is due to the compressing of image space.

**Strengths:**

- The observation of the increasing $\ell_p$ distance to the clean sample produced by diffusion-based purification is interesting.
- The definition of fuzzy adversarial robustness is meaningful and the proposed CFR curve can be a practical evaluation metric for stochastic defense methods.
- The figures clearly demonstrate the ideas of the paper.

**Weaknesses:**

- The description of the previous theoretical result in (Nie et al., 2022) is unfaithful.
  - In Section 3.1, it is stated that the previous explanation of the robustness of diffusion-based purification methods lies in the "shrinkage of the adversarial attack", or more specifically, the decreasing $\ell_p$ distance to the clean sample after purification.
  - Nonetheless, Theorem 3.1 in (Nie et al., 2022) only suggests that the divergence between the clean data distribution and adversarial sample distribution can be decreased by the *forward diffusion process*. It is not assumed that the purified sample obtained by the *diffusion-denoising purification process* is closer to the clean sample under $\ell_p$ distance.
  - From my perspective, while the purified sample is expected to lie on the non-adversarial manifold, its $\ell_p$ distance to the clean sample is not important as long as the semantic information is preserved.
- The hyperspherical cap model is not well supported.
  - Crossing the critical threshold along an adversarial direction can be a necessary condition for adversarial samples, but it is insufficient. Specifically, Assumption 2 is valid locally given the continuity of the model, but there can be an upper bound on radius $\hat{\gamma}(x_0, \eta)$ determined by $x_0$ and the direction $\eta$ for this assumption to hold. In other words, the sample may cross the boundary again as the radius exceeds $\hat{\gamma}(x_0, \eta)$. An example of such a decision boundary is depicted in Figure 1 of [1].
  - Therefore, unless a non-trivial uniform upper bound $\hat{\gamma}$ is derived, it cannot be claimed that crossing the critical threshold is a sufficient condition for adversarial examples within an $\ell_2$ neighborhood with a certain radius.
  - It is also assumed that "the classification boundaries are locally linear" (Lines 375-376), which is not supported by valid theoretical or empirical evidence. While the decision boundaries depicted in the 2D projection in Figure 4(b,c) appear to be linear, they are likely non-linear in the high-dimensional input space. Stronger evidence or proper references are required to claim this point.
- The causal relationship between the compression of image space and the improved robustness is not well explained.
  - Section 6 has validated that diffusion-based purification can compress the image space, but it's still not apparent how it contributes to the robustness. For example, while the magnitude of the critical threshold can be reduced due to the compression, the purified sample $f(x_0)$ may also lie closer to the decision boundary, which cannot explain the improved robustness.
  - It is better to clarify whether the compression effect is the sole contributor to the robustness of diffusion-based purification models.
- The little-o notation in Line 717 is inappropriate since two real numbers are compared here instead of two growing functions. The "considerably larger" change of logit should be better defined.

[1] Kim, Hoki, Woojin Lee, and Jaewook Lee. "Understanding catastrophic overfitting in single-step adversarial training." AAAI 2021.

**Questions:**

- If image space compressing is sufficient for improving the robustness, are conventional image compressing methods like JPEG effective for adversarial defense?

---

> ### Author Response · Authors · 2024-11-30
>
> We thank the reviewer for the thoughtful feedback and appreciate the opportunity to address the concerns raised. Below, we provide detailed responses to the key weaknesses identified.
>
> ### **Replies to Weaknesses**
>
> #### **Clarification on the Misclaim**
> - We would like to clarify regarding the misclaim about prior explanations of robustness improvement under diffusion models, which was also highlighted by reviewer **aGTV**. Our intent was to highlight that prior works offered intuitive explanations, which *motivated* us to investigate whether diffusion models increase or decrease $\ell_p$ norms during adversarial purification. This question remains under-investigated to the best of our knowledge.
>
>   We agreed that prior works did not claim, nor experimentally demonstrate, that “diffusion models improve robustness by decreasing $\ell_p$ norms to clean images.” To ensure clarity, we plan to rephrase **Sec. 3.1** as follows:
>
>   > “While the exact mechanisms for robustness improvement under diffusion models remain unclear, intuitive explanations have been discussed in the DiffPure paper [Nie et al., 2022], e.g., diffusion models ``recover clean images through the reverse denoising process.'' This motivates us to test a simple hypothesis: diffusion models shrink the $\ell_p$ distances towards clean images during adversarial purification. ”
>
>   We thank both reviewers for pointing out this ambiguity. If there are further suggestions for refining this section, we welcome your input. We believe this revision will effectively address the concern while preserving the integrity of our experiments and main arguments.
>
>
> - Regarding the theoretical explanation in Theorem 3.1 of DiffPure [1], we appreciate the reviewer drawing attention to its significance. However, we believe it does not fully explain how diffusion models enhance adversarial robustness. Specifically, the theorem addresses the forward process and shows that adding Gaussian noise reduces the $\ell_p$ distances between two distributions. However, it does not account for the fact that diffusion models with only the **reverse denoising process** can also improve robustness (line 172 of our paper, a finding also reported in DensePure [2]). Additional experimental results highlighting this phenomenon will be included in the Appendix, as suggested by reviewers **7QBf** and **iBpK**.
>
>   Our proposed explanations of the **adversarial compression effect** and the **hyperspherical cap model** provide alternative perspectives on the reverse denoising process and its role in adversarial robustness. We view these results as complementary insights of the original theoretical work developed in the DiffPure paper.
>
> - In response to suggestions from reviewers **7QBf** and **cqB1**, we evaluated the **structural similarity index measure (SSIM)** [2], a perceptual metric, in addition to $\ell_p$ distances. The results are summarized below:
>
>   | Distance (to clean images)         | Adversarial (PGD-EOT, $\ell_\infty=8/255$) | Random (uniform, $\ell_\infty=8/255$)       | Purified states        |
>   |-------------------------------------|------------------------------------------------------------|-----------------------------------------------------------|----------------------|
>   | **SSIM**                            | 0.963 $\pm$ 0.030                                             | 0.966 $\pm$ 0.032                                          | 0.791 $\pm$ 0.085   |
>
>   As shown, we observed an approximately 20% decrease in SSIM after diffusion purification. This interesting complementary finding will be incorporated into the final version of the paper.
>
>   However, we continue to emphasize the importance of $\ell_p$ distance measurements in the context of adversarial purification. This is because adversarial attacks are inherently constructed based on $\ell_p$ distances. Diffusion models do not merely convert adversarial perturbations into smaller $\ell_p$ distances, which would simplify the problem; instead, they purify states, making them **perceptually closer** to the original images while potentially increasing $\ell_p$ distances. This highlights a unique aspect of their operation that we believe warrants further exploration.

---

> ### Author Response · Authors · 2024-11-30
>
> #### **The Hyperspherical Cap Model**
> - We appreciate the reviewer’s concern. For the classifier we studied (WideResNet-28-10, standard classifier from RobustBench [4]) on CIFAR-10, we did not observe multiple crossings of the decision boundary along the adversarial direction within the typical scale of adversarial perturbations. Specifically, we observed such effects (e.g., flipping to other classes) only when the $\ell_2$ norm exceeded 5—much larger than the radius of adversarial examples typically considered (e.g., $\ell_\infty=8/255$, roughly corresponding to $\ell_2=1$). Most class vs. $\ell_2$ distance curves along adversarial directions were step functions, indicating no transitions or at most a single transition within the adversarial ball.
>
>     We consider this to be an interesting result. Indeed, one might have expected a different outcome before these experiments were done. We will formally quantify these findings in the final version to further support the validity of Assumption 2.
>
> - The above observations should address the reviewer’s concern. If OpenReview permits, we are happy to share the class vs. $\ell_2$ curves directly. Alternatively, we welcome suggestions on specific metrics the reviewer considers important. These results are reproducible, and we will include details of our experimental settings in the paper (WideResNet-28-10, "Standard" $\ell_\infty$ classifier from RobustBench on CIFAR-10, scanning along PGD attacks up to $\ell_2=1$).
>
> - We acknowledge the difficulty of sampling points around adversarial directions in high-dimensional spaces due to their sparsity. Inspired by [5], we projected adversarial directions onto 2D planes defined by random directions to study the loss landscape. To address this concern, we refined our methods by sampling multiple 2D slices. Instead of one 2D slice per image, we now use 100 slices, with the same 1000 points per slice. This new method allows us to sample high-dimensional space around the adversarial direction, addressing a key limitation of the previous method we used that was pointed out by the reviewer.
>
>     This refinement slightly changed the estimated slope of the psychometric function, with $\bar{k} = 5.9106 \pm 1.0737$. Importantly, the new result still supports our claim that the transition is sharp. We will include this new analysis in the revised manuscript. We believe the refined analysis makes this point stronger. We thank you for this suggestion.
>
>
> #### **Adversarial Compression**
>
> - We clarify that the critical threshold (Assumption 2) reflects the distance to decision boundaries along adversarial directions and is an inherent property of the classifier. The purification system (e.g., diffusion models) does not alter this threshold. Instead, purification compresses the distances of adversarial examples toward the anchor point $f(x_0)$, preventing them from crossing the threshold. Two factors influence robust/non-robust outcomes:
>        1. The amount of compression induced by the diffusion model for each image.
>        2. The critical threshold of the purified clean image $f(x_0)$.
>     As illustrated in Fig. 6, increased compression moves samples further from decision boundaries, leading to improved robustness. The reviewer’s intuition is correct that distances to the decision boundary at $f(x_0)$ significantly affect robustness outcomes.
>
> - We emphasize that our paper identifies two critical effects:
>          1. **Adversarial compression rate** (a property of the purification system).
>          2. **Critical threshold** (a property of the classification system).
>         As shown in Fig. 6, both effects strongly correlate with robustness outcomes, and robustness is not solely determined by compression.
>
> #### **Notations**
> - We thank the reviewer for pointing out the notation issue. To clarify, we propose replacing the small-o notation with the "much less than" symbol ($\lll$). By "considerably large," we refer to changes in logits along random projections that are negligible compared to changes along adversarial directions. We hope this adjustment makes our argument clearer.

---

> ### Author Response · Authors · 2024-11-30
>
> ### **Replies to Questions**
>
> - Compression in JPEG vs. Diffusion Models
>
>   We acknowledge the potential confusion between "compression" in our context and in JPEG. In JPEG, compression refers to reducing storage bits, often by applying a discrete cosine transform and removing high-frequency components, effectively acting as a low-pass filter. In contrast, our use of "compression" refers to shrinking distances in the image space toward purified clean images $f(x_0)$.
>
>   Regarding the potential robustness of JPEG compression, previous studies [6] suggest that adversarial training biases models toward low-frequency information, which may explain why JPEG could improve robustness. However, we are more interested in studying robustness as an emergent property of learning systems (e.g., denoising or diffusion models) rather than from engineered constraints like low-pass filters or JPEG compression.
>
> We hope our responses address the reviewer’s concerns satisfactorily. Please feel free to provide further feedback or raise additional questions.
>
> ---
> ### **References**
> [1] Nie, W., Guo, B., Huang, Y., Xiao, C., Vahdat, A., & Anandkumar, A. (2022, June). Diffusion Models for Adversarial Purification. In International Conference on Machine Learning (pp. 16805-16827). PMLR.
>
> [2] Xiao, C., Chen, Z., Jin, K., Wang, J., Nie, W., Liu, M., ... & Song, D. (2023). Densepure: Understanding diffusion models for adversarial robustness. In The Eleventh International Conference on Learning Representations.
>
> [3] Wang, Z., Bovik, A. C., Sheikh, H. R., & Simoncelli, E. P. (2004). Image quality assessment: from error visibility to structural similarity. IEEE transactions on image processing, 13(4), 600-612.
>
> [4] Croce, F., Andriushchenko, M., Sehwag, V., Debenedetti, E., Flammarion, N., Chiang, M., ... & Hein, M. (2020). Robustbench: a standardized adversarial robustness benchmark. arXiv preprint arXiv:2010.09670.
>
> [5] Li, H., Xu, Z., Taylor, G., Studer, C., & Goldstein, T. (2018). Visualizing the loss landscape of neural nets. Advances in neural information processing systems, 31.
>
> [6] Yin, D., Gontijo Lopes, R., Shlens, J., Cubuk, E. D., & Gilmer, J. (2019). A fourier perspective on model robustness in computer vision. Advances in Neural Information Processing Systems, 32.

---

> > ### Comment · Reviewer_aGTV · 2024-12-01
> >
> > I thank the authors for their thorough responses to my review. Here are my replies.
> >
> > - **Clarification on the Misclaim**: I accept the clarification on the misclaim about prior explanations of how diffusion-based purification methods achieve robustness. I acknowledge that this issue can be addressed by moderate revision of the relevant texts in this paper, without significantly changing the related conclusions and contributions. The supplementary experiments in the rebuttal on perceptual distance further complement the understanding of the behavior of these methods.
> > - **The Hyperspherical Cap Model**: I appreciate the authors' attempt to formalize concepts like "adversarial directions" and propose the hyperspherical cap model to depict the adversarial regions, and my major concern is the soundness of these theoretical models. The authors' responses have partly addressed this concern, e.g., by empirically showing that the decision boundary is not likely to be crossed a second time along an adversarial direction within the common perturbation ranges in existing studies. However, as the theoretical models are grounded on limited empirical observations, the significance and contribution of this point may be controversial.
> > - **Adversarial Compression**:
> >   - I previously thought that the critical threshold here is the property of the whole purification-classification system. Now I understand the claims related to Fig. 6, where the compression effect of the purification model refers to reducing "the distances of adversarial examples toward the anchor point $f(x _ 0)$", as explained in the rebuttal. I acknowledge that the disentanglement of the compression rate and the critical distance provides a new perspective on the robustness of purification-based defense.
> >   - I have read the authors' response to Reviewer 7QBf and pJXW on the possible future directions motivated by the findings and perspectives of this paper. However, according to Fig. 6, an important direction is to ensure that the critical threshold for $f(x _ 0)$ is large enough, but this is not explicitly mentioned by the authors.
> > - An additional concern is that the fuzzy robustness defined in Sec. 4 and the stochastic nature of diffusion models seem not considered in the discussions in Sec. 5 and Sec. 6. Will stochasticity affect these discussions?
> >
> > Overall, I appreciate the contributions of empirical studies on the intriguing behavior of diffusion models (Sec. 3) and the concept of fuzzy robustness (Sec. 4). The discussions in Sec. 5 and Sec. 6 are intuitive but not rigorous enough, and their significance on understanding and improving diffusion-based purification methods seem to be limited.
> >
> > **Therefore, I will increase my score from 3 to 5 for now, and I look forward to further discussions with the authors and other reviewers.**

---

> > > ### Author Response · Authors · 2024-12-04
> > >
> > > We appreciate the reviewer’s continued engagement and insightful questions, which have greatly enriched the discussion. Below, we provide further clarifications to address the concerns raised:
> > >
> > > - **Critical Thresholds at $f(x_0)$**
> > > We seek to clarify the reference to “the critical threshold for $ f(x_0) $ is large enough.” Indeed, the diffusion model does not substantially alter the critical threshold.
> > >
> > > There are four critical thresholds to consider along adversarial directions:
> > >   1. The worst (PGD) of the classifier at $ x_0 $.
> > >   2. The worst of the classifier at $ f(x_0) $.
> > >   3. The attack threshold for the entire system at $ x_0 $ (before purification).
> > >   4. The attack threshold for the entire system at $ f(x_0) $ (after purification).
> > >
> > > All four thresholds are positively correlated, meaning samples with smaller critical thresholds typically lie closer to decision boundaries. Moreover, thresholds (3) and (4) are similar in scale, suggesting that the diffusion model tends to move samples along the direction of the decision boundary rather than orthogonal to it.
> > > We will provide a detailed quantification of this effect in the final version, which we hope will further address your question.
> > >
> > > - **Stochastic and Adversarial Compression**
> > > Yes, your intuition aligns with our findings. Introducing stochasticity complicates both the conceptual discussion and computational analysis. Thus, as stated at the beginning of Section 6, we focus on studying the compression effect within a specific randomness configuration:
> > >
> > > > “Next, we seek to understand how the diffusion models improve robustness in adversarial purification **within a particular randomness configuration**.”
> > >
> > > Under this assumption, the diffusion model effectively operates as a deterministic mapping, as illustrated in Figure 2c. This approach simplifies the analysis while preserving the core findings related to compression.
> > >
> > > Please let us know if further clarifications or data would be helpful. We welcome continued feedback and are grateful for this constructive dialogue.

---

### Meta-Review · Area_Chair_PaGw · 2024-12-20

**Metareview:**

This paper examines how diffusion models improve adversarial robustness, challenging the idea that these models bring adversarial samples closer to clean images. Instead, the authors argue that purified images actually move farther away in $\ell_p$  space, and that the robustness comes from image space compression and the randomness inherent to diffusion models. They introduce "fuzzy adversarial robustness" to account for the stochastic nature of these models and propose a hyperspherical cap model to explain adversarial regions. These are novel contributions backed by empirical evidence, with an intriguing take on how diffusion-based defenses function.

Nevertheless, there are some noticable weaknesses. Reviewers aGTV is particularly concerned about the theoretical claims—like the hyperspherical cap model—which lack strong mathematical or empirical backing. Also, the fuzzy robustness concept overlaps with prior work on randomized smoothing. Another issue is the limited exploration of practical applications.

Based on these points, I concur with most reviewers and recommend rejecting the paper for now. I encourage the authors to consider these comments and revise the paper accordingly for a future venue.

**Additional Comments On Reviewer Discussion:**

The reviewers had a lot of back-and-forth about this paper. On the plus side, they agreed that the observations—like purified images moving farther from clean ones—are novel and worth exploring. Reviewer aGTV raised valid concerns about the hyperspherical cap model and whether it’s supported by evidence. Reviewer 7QBf appreciated the analysis but pointed out gaps, like not comparing denoiser-based defenses. Reviewer cqB1 questioned the limited scope of sampling methods and metrics used, while iBpK noted ambiguities in the figures and flagged overlaps with prior concepts.

The authors responded thoroughly, clarifying many points and adding new experiments, like using perceptual metrics. While some reviewers were satisfied with these efforts and raised their scores, others felt the core issues—like the lack of theoretical rigor and the practical utility of the findings—remained unresolved. Ultimately, while the rebuttal helped clarify certain aspects, the paper’s contributions still feel incomplete. It’s a solid step forward but not quite ready for publication.

---

### Decision · Program_Chairs · 2025-01-22

Reject